# Addressing Hidden Confounding with Heterogeneous Observational Datasets for Recommendation

**Yanghao Xiao**[1,3]    **Haoxuan Li**[2]    **Yongqiang Tang**[3,*]    **Wensheng Zhang**[4,*]

[1]University of Chinese Academy of Sciences    [2]Peking University
[3]Institute of Automation, Chinese Academy of Sciences    [4]Guangzhou University
`xiaoyanghao22@mails.ucas.ac.cn, hxli@stu.pku.edu.cn,`
`yongqiang.tang@ia.ac.cn, zhangwenshengia@hotmail.com`

## Abstract

The collected data in recommender systems generally suffers selection bias. Considerable works are proposed to address selection bias induced by observed user and item features, but they fail when hidden features (e.g., user age or salary) that affect both user selection mechanism and feedback exist, which is called hidden confounding. To tackle this issue, methods based on sensitivity analysis and leveraging a few randomized controlled trial (RCT) data for model calibration are proposed. However, the former relies on strong assumptions of hidden confounding strength, whereas the latter relies on the expensive RCT data, thereby limiting their applicability in real-world scenarios. In this paper, we propose to employ heterogeneous observational data to address hidden confounding, wherein some data is subject to hidden confounding while the remaining is not. We argue that such setup is more aligned with practical scenarios, especially when some users do not have complete personal information (thus assumed with hidden confounding), while others do have (thus assumed without hidden confounding). To achieve unbiased learning, we propose a novel meta-learning based debiasing method called MetaDebias. This method explicitly models oracle error imputation and hidden confounding bias, and utilizes bi-level optimization for model training. Extensive experiments on three public datasets validate our method achieves state-of-the-art performance in the presence of hidden confounding, regardless of RCT data availability.

## 1 Introduction

Recommender systems have been developed with the purpose of providing users with personalized recommendations, serving as a potent tool in capturing users' true preferences [22]. In recent years, deep learning algorithms are proposed to train recommendation models with collected historical data. However, the selection bias introduced by the users' selective interactions with items poses a challenge to the unbiased learning of the prediction model [4, 54, 59]. For instance, in explicit feedback data, users are free to rate items they prefer, thereby engendering the distribution discrepancy between interacted and non-interacted data [35, 44, 55].

To alleviate selection bias, a line of works are proposed, including error imputation [2, 16, 48], inverse propensity weighting [21, 36, 45, 46], and doubly robust [39, 43, 55]. Moreover, recent works have sought to enhance these foundational methods from diverse perspectives, including analysis on both bias and variance [7, 12, 27], considerations for training stability [31, 49], and integration with multi-task learning [37, 52, 64]. However, they all fail to achieve unbiased when some features

---

*Corresponding author

38th Conference on Neural Information Processing Systems (NeurIPS 2024).

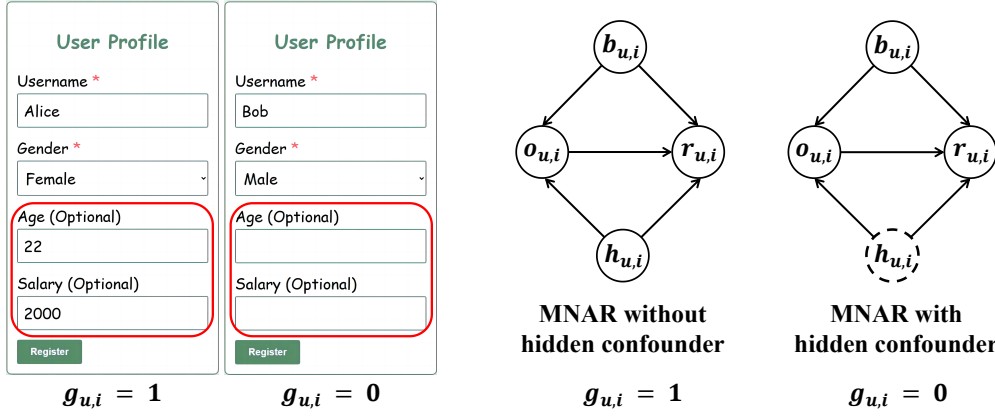

(a) Toy examples of user information collection process.     (b) The causal graphs of two types of MNAR data.

Figure 1: Toy examples and causal graphs of heterogeneous observational data, both of which are missing not at random (MNAR) due to the selection bias. In causal graphs, $b_{u,i}$, $o_{u,i}$, $r_{u,i}$ and $h_{u,i}$ denote basic mandatory features, observation, rating and optional features, respectively, where observed and unobserved variables are represented by solid-line and dashed-line circles.

that simultaneously affect both user selection mechanism and feedback remain unobserved, which is called hidden confounding and is widely prevalent in real-world scenarios [10, 28, 29].

To address hidden confounding, sensitivity analysis based methods assume the hidden confounding strength can be bounded such that the inverse of true propensity scores are near and bounded by the estimated values, and further adopt worst-case optimization [10, 66]. More recently, data fusion based methods leverage a few RCT data collected from randomized controlled trials or A/B tests to calibrate propensity and imputation models to achieve unbiasedness [28, 29]. Unfortunately, both of them prove challenging to implement in real-world scenarios, as the former relies on strong assumption on hidden confounding strength, whereas the latter relies on the significantly expensive RCT data.

To fill this gap, this paper proposes to employ heterogeneous observational data to address hidden confounding, wherein some data is affected by hidden confounding while the remaining is not. We argue that such setup is more aligned with practical recommendation scenarios, especially when some users/items do not have complete information (thus assumed with hidden confounding), while others do have (thus assumed without hidden confounding). For example, in Figure 1a, during user registration process, some users such as Alice may complete both mandatory fields and optional fields like age and salary, while others such as Bob may only complete mandatory fields. Similarly, this also holds on the item side. We depict the corresponding causal graphs in Figure 1b, where $b_{u,i}$ and $h_{u,i}$ denote basic mandatory and optional features that affect both observation and feedback, respectively. To achieve unbiased learning with above heterogeneous observational data, we propose a novel meta-learning based debiasing method called MetaDebias, to explicitly model the oracle prediction errors and the bias introduced by hidden confounders. We further adopt bi-level optimization technique to train the prediction model. The contributions of this work are summarized as follows.

• To the best of our knowledge, this is the first work to employ heterogeneous observational data to address hidden confounding in debiased recommendation, relaxing the reliance on RCT data in previous data fusion methods.

• We propose a meta-learning based debiasing method called MetaDebias to explicitly model the oracle error imputation, and employ bi-level optimization for model training.

• We conduct extensive experiments on three public datasets, and our method achieves state-of-the-art performance in the presence of hidden confounding, regardless of the availability of RCT data.

## 2   Preliminaries and Related Work

Let $\mathcal{U} = \{u\}$, $\mathcal{I} = \{i\}$, and $\mathcal{D} = \{(u,i) \mid u \in \mathcal{U}, i \in \mathcal{I}\}$ be the set of users, the set of items, and the target population consisting of all user-item pairs, respectively. Define $x_{u,i}$ and $r_{u,i}$ be the observed features and rating of user-item pair $(u,i)$, and suppose observation indicator $o_{u,i} \in \{0,1\}$

be a binary treatment variable, where $o_{u,i} = 1$ indicates $r_{u,i}$ is observed, otherwise is not. Let $\mathcal{O} = \{(u,i) \mid (u,i) \in \mathcal{D}, o_{u,i} = 1\}$ be the observed population consisting of user-item pairs with observed ratings. Denote $\mathbb{P}$ and $\mathbb{E}$ be the distribution and expectation on the target population $\mathcal{D}$.

We denote prediction model with parameter $\theta$ as $\hat{r}_{u,i} = f(x_{u,i}; \theta)$ which aims to predict true ratings. If all the ratings $\{r_{u,i} : (u,i) \in \mathcal{D}\}$ are observed, i.e., $\mathcal{O} = \mathcal{D}$, the parameter $\theta$ can be trained by minimizing the ideal loss:

$$\mathcal{L}_{\text{ideal}}(\theta) = \frac{1}{|\mathcal{D}|} \sum_{(u,i) \in \mathcal{D}} e_{u,i},$$

where $e_{u,i} = L(\hat{r}_{u,i}, r_{u,i})$ is the prediction error and $L(\cdot, \cdot)$ is a pre-specified loss function such as the squared loss $e_{u,i} = (\hat{r}_{u,i} - r_{u,i})^2$. However, rating $r_{u,i}$ is observed only when $o_{u,i} = 1$, and naively minimizing prediction loss on the observed population $\mathcal{O}$ will suffer from bias and lead to sub-optimal performance [55]. This is attributed to the disparities between the observed population $\mathcal{O}$ and the target population $\mathcal{D}$, which is called selection bias.

## 2.1 Addressing Selection Bias without Hidden Confounding

To address selection bias, prior works preliminarily assume the absence of hidden confounding in the collected data, implying that the observed features $x_{u,i}$ include all the possible confounders. Without loss of generality, we define a binary variable $g_{u,i} \in \{0,1\}$ as the data source indicator which measures the completeness of observed features $x_{u,i}$. As shown in the toy example in Figure 1, $g_{u,i} = 1$ indicates that all confounders are completely observed, while $g_{u,i} = 0$ indicates that some hidden confounders exist. To ensure consistency in the input, we fill the missing values in $x_{u,i}$ with 0, so that the observed features $x_{u,i}$ have the same dimensions when $g_{u,i} = 0$ and $g_{u,i} = 1$.

Assumed no hidden confounding ($g_{u,i} = 1, (u,i) \in \mathcal{D}$), it holds that $e_{u,i} \perp\!\!\!\perp o_{u,i} \mid x_{u,i}, g_{u,i} = 1$, the widely adopted propensity based methods are proposed. Specifically, the Inverse Propensity Scoring (IPS) estimator [46] reweights each sample based on the probability of being observed and is given as

$$\mathcal{L}_{\text{IPS}}(\theta) = \frac{1}{|\mathcal{D}|} \sum_{(u,i) \in \mathcal{D}} \frac{o_{u,i} e_{u,i}}{\hat{p}_{u,i}},$$

where $\hat{p}_{u,i}$ is the estimation of propensity $p_{u,i} \triangleq \mathbb{P}(o_{u,i} = 1 | x_{u,i}, g_{u,i} = 1)$. Furthermore, the Doubly Robust (DR) estimator [43, 55] further extends IPS with error imputation and is presented as

$$\mathcal{L}_{\text{DR}}(\theta) = \frac{1}{|\mathcal{D}|} \sum_{(u,i) \in \mathcal{D}} [\hat{e}_{u,i} + \frac{o_{u,i}(e_{u,i} - \hat{e}_{u,i})}{\hat{p}_{u,i}}],$$

where $\hat{e}_{u,i}$ is the imputed error. Based on IPS and DR estimators, many variants have been proposed, and please refer to Appendix A.1 for more details.

## 2.2 Addressing Selection Bias with Hidden Confounding

However, in some real-world scenarios, the observed features do not include all confounders and hidden confounding exists. Based on the incomplete observed features $x_{u,i}$ with $g_{u,i} = 0$, it holds that $o_{u,i} \not\perp\!\!\!\perp e_{u,i} \mid x_{u,i}, g_{u,i} = 0$, which prevents previous methods such as DR from achieving unbiasedness. To further mitigate the hidden confounding, methods based on sensitivity analysis and model calibration using RCT data have been proposed.

**Sensitivity Analysis.** Inspired from causal inference literature, sensitivity analysis based approach assumes the hidden confounding strength can be bounded such that the inverse of true propensity scores are near and bounded by the estimated values, and adopt worst-case optimization to mitigate hidden confounding [10, 66]. However, above strong assumption on hidden confounding strength is hard to be satisfied in real-world scenarios and such method fails when the assumption is violated.

**Model Calibration.** Recent works propose to leverage a few unbiased RCT data collected from randomized controlled trials or A/B tests for model calibration [28, 29]. Collecting RCT data requires users to rate items randomly, hence RCT samples are representatives of the target population, and the prediction loss on these samples serves as an unbiased estimator of the ideal loss. Thus, the biased propensity and imputation models learned from the incomplete observed features $x_{u,i}$ can be

corrected using such unbiased loss, for instance, with the help of additive residual models [28] or multiplicative reweighting models [29]. However, the acquisition cost of RCT data is prohibitively high, posing challenges to the practical implementation of such methods in real-world settings.

Apart from debiased recommendation, more related works about causal inference under hidden confounding can be found in Appendix A.1, which is a widely studied topic.

# 3 MetaDebias: Meta Learning Based Debiased Recommendation Approach

## 3.1 Problem Formulation

We first present the problem formulation, that is to provide an unbiased estimator for the ideal loss $\mathcal{L}_{\text{ideal}}$ in the presence of hidden confounding, given the collected heterogeneous observational data with known data source indicator $g_{u,i}$. Unlike existing works that assume training data originates from a single dataset without or with hidden confounding, we argue that the training data is more likely composed of two heterogeneous datasets, one of which has sufficient feature collection and is assumed without hidden confounding ($g_{u,i} = 1$), while the other has insufficient feature collection and is assumed with hidden confounding ($g_{u,i} = 0$) with hidden confounders denoted as $h_{u,i}$.

Here, based on the given $g_{u,i} \in \{0, 1\}$ for distinguishing between two types of observational data, we partition the observed population $\mathcal{O}$ into two subpopulations $\mathcal{D}_0 = \{(u,i) \mid (u,i) \in \mathcal{D}, g_{u,i} = 0, o_{u,i} = 1\}$ and $\mathcal{D}_1 = \{(u,i) \mid (u,i) \in \mathcal{D}, g_{u,i} = 1, o_{u,i} = 1\}$, while unobserved population is $\mathcal{D}_u = \{(u,i) \mid (u,i) \in \mathcal{D}, o_{u,i} = 0\}$. A naive approximation method for ideal loss is minimizing the prediction loss over the observed data. Specifically, given $\mathcal{D}_0$ and $\mathcal{D}_1$, the respective losses over each subgroup are as follows:

$$\mathcal{L}_{\mathcal{D}_0} = \frac{1}{|\mathcal{D}_0|} \sum_{(u,i) \in \mathcal{D}_0} e_{u,i}, \qquad \mathcal{L}_{\mathcal{D}_1} = \frac{1}{|\mathcal{D}_1|} \sum_{(u,i) \in \mathcal{D}_1} e_{u,i}.$$

However, due to the selection bias, neither $\mathcal{L}_{\mathcal{D}_0}$, $\mathcal{L}_{\mathcal{D}_1}$ nor their combination $\mathcal{L}_{\text{Naive}} = \mathcal{L}_{\mathcal{D}_0} + \mathcal{L}_{\mathcal{D}_1}$ is an unbiased estimator for the ideal loss. Furthermore, as discussed in the Preliminaries 2, previous methods cannot be applied to achieve the goal without additional hidden confounding strength or RCT data assumptions.

## 3.2 Methodology

To achieve the goal, we propose to explicitly estimate the prediction error $e_{u,i}$ on all user-item pairs $\mathcal{D}$ using observed features $x_{u,i}$, as the ideal loss is defined as the average prediction error over $\mathcal{D}$, i.e., $\mathcal{L}_{\text{ideal}} = \frac{1}{|\mathcal{D}|} \sum_{(u,i) \in \mathcal{D}} e_{u,i}$. In other words, our goal is transformed into accurately estimating the oracle error imputation $\mathbb{E}[e_{u,i} \mid x_{u,i}]$, which can inherently derive an unbiased estimator.

The challenges in achieving accurate oracle error imputation estimation lie in the fact that the prediction error $e_{u,i}$ is only partially observable, and the missing mechanisms differ between the two subgroups $\mathcal{D}_0$ and $\mathcal{D}_1$ due to the influence of hidden confounding. To address these challenges, we define the identifiable propensity scores aimed at modeling the two types of missing mechanisms, along with the naive error imputations defined over the entire space $\mathcal{D}$, to assist in estimating the target oracle error imputation. Next, we will introduce the details.

To start with, different from the previously widely used propensity score $\mathbb{P}(o_{u,i} = 1 | x_{u,i})$ which is applicable only for samples in subgroup $\mathcal{D}_1$ and the generalized formulation $\mathbb{P}(o_{u,i} = 1 | x_{u,i}, h_{u,i})$ designed for subgroup $\mathcal{D}_0$ which is unidentifiable, we define the identifiable propensity score $\pi(x, g)$ which models the probability of being observed for any user-item pair on the entire space $\mathcal{D}$:

$$\pi(x, g) = \mathbb{P}\left(o_{u,i} = 1 \mid x_{u,i} = x, g_{u,i} = g\right). \tag{1}$$

The proposed $\pi(x, g)$ models the propensity score for both the absence and presence of hidden confounding, based on the input data source indicator $g_{u,i}$.

Similarly, we propose to use the observed features $x_{u,i}$ and data source indicator $g_{u,i}$ to estimate the naive prediction error as $o_{u,i} \cdot e_{u,i}$, which is computable on the target population $\mathcal{D}$ and expected to be the prediction error $e_{u,i}$ when sample $(u, i)$ is observed $o_{u,i} = 1$, or to be zero when not

observed $o_{u,i} = 0$. Note that the zero value in the defined naive prediction error $o_{u,i} \cdot e_{u,i}$ contains the information about data missing mechanism, thus $g_{u,i}$ is used to capture the differences in the missing mechanism when hidden confounding is present or absent. Formally, to achieve this target, we define the naive error imputation $m(x, g)$ as:

$$m(x, g) = \mathbb{E}[o_{u,i} \cdot e_{u,i} \mid x_{u,i} = x, g_{u,i} = g]. \tag{2}$$

Based on the proposed propensity score $\pi(x, g)$ and naive error imputation $m(x, g)$, the oracle error imputation $\mathbb{E}[e_{u,i} \mid x_{u,i}]$ can be derived, we summarize their relationship in Lemma 1 below.

**Lemma 1.** *The relationship between the proposed propensity score $\pi(x, g)$, naive error imputation $m(x, g)$ and the oracle error imputation $\mathbb{E}[e_{u,i} \mid x_{u,i}]$ is as follows:*

$$m(x, g) = \{\mathbb{E}[e_{u,i} \mid x_{u,i} = x] + (1 - g)\eta(x)\} \cdot \pi(x, g), \tag{3}$$

*where $\eta(x) = \mathbb{E}[e_{u,i} \mid x_{u,i} = x, g_{u,i} = 0, o_{u,i} = 1] - \mathbb{E}[e_{u,i} \mid x_{u,i} = x]$.*

The above result delineates the conditional independence relationship between observation and prediction error. Specifically, on subgroup $\mathcal{D}_1$ where no hidden confounding exists, the prediction error $e_{u,i}$ is independent of the observation $o_{u,i}$ given the observed features $x_{u,i}$, i.e., $e_{u,i} \perp\!\!\!\perp o_{u,i} \mid x_{u,i}, g_{u,i} = 1$, and the expectation of $o_{u,i} \cdot e_{u,i}$ naturally equals the product of their respective expectations as shown in Equation 3.

However, on subgroup $\mathcal{D}_0$ where hidden confounders $h_{u,i}$ exist, above conditional independence does not hold, thus we introduce an additional residual module $\eta(x)$ which describes the expectation difference between the mean prediction error over target population $\mathcal{D}$ and the biased subpopulation $\mathcal{D}_0$, to account for the bias introduced by hidden confounders.

Lemma 1 provides an approach for estimating the oracle error imputation $\mathbb{E}[e_{u,i} \mid x_{u,i}]$, and the estimation depends solely on propensity score $\pi(x, g)$ and naive error imputation $m(x, g)$. Note that both $\pi(x, g)$ and $m(x, g)$ are typically estimated from the heterogeneous observational datasets, which implies that when the estimation of $\pi(x, g)$ and $m(x, g)$ are inaccurate, the resultant oracle error imputation estimation will further deteriorate in accuracy.

To address this limitation, we propose further incorporating additional data information including the observation $o_{u,i}$ and the naive prediction error $o_{u,i} \cdot e_{u,i}$ to achieve a more robust estimation of the oracle error imputation. The details are shown in Lemma 2 below.

**Lemma 2.** *The relationship between the propensity score $\pi(x, g)$, naive error imputation $m(x, g)$, observation $o_{u,i}$, naive prediction error $o_{u,i} \cdot e_{u,i}$ and oracle error imputation $\mathbb{E}[e_{u,i} \mid x_{u,i}]$ is:*

$$o_{u,i} \cdot e_{u,i} - m(x, g) = \{\mathbb{E}[e_{u,i} \mid x_{u,i} = x] + (1 - g)\eta(x)\} \cdot \{o_{u,i} - \pi(x, g)\} + \xi, \tag{4}$$

*where $\xi = o_{u,i} \cdot \{e_{u,i} - \{\mathbb{E}[e_{u,i} \mid x_{u,i} = x] + (1 - g)\eta(x)\}\}$ with $\mathbb{E}[\xi \mid x, g] = 0$.*

The findings in Lemma 2 provide an alternative and robust approach for estimating the oracle error imputation $\mathbb{E}[e_{u,i} \mid x_{u,i}]$, wherein the estimation depends not only on the learned propensity score $\hat{\pi}(x, g)$, naive error imputation $\hat{m}(x, g)$, but also on the observation $o_{u,i}$, along with the naive prediction error $o_{u,i} \cdot e_{u,i}$, where $\xi$ with zero-mean property can be regarded as a noise.

### 3.3 Training Objectives

For the method implementation, we employ deep learning models to estimate the aforementioned proposed modules, and name this approach as MetaDebias. We present the architecture of MetaDebias in Figure 2, and introduce the training objective of each model as follows.

Initially, we employ the commonly used cross-entropy loss to train the propensity model $\pi(x, g; \phi_\pi)$ with parameters $\phi_\pi$ using heterogeneous data, denoted as $\mathcal{L}_\pi(\phi_\pi)$ shown in Equation 5 below:

$$\mathcal{L}_\pi(\phi_\pi) = \frac{1}{|\mathcal{D}|} \sum_{(u,i) \in \mathcal{D}} [-o_{u,i} \cdot \log \pi(x_{u,i}, g_{u,i}) - (1 - o_{u,i}) \cdot \log(1 - \pi(x_{u,i}, g_{u,i}))]$$

$$= \frac{1}{|\mathcal{D}|} \sum_{(u,i) \in \mathcal{D}_0 \cup \mathcal{D}_1} -\log \pi(x_{u,i}, g_{u,i}) + \frac{1}{|\mathcal{D}|} \sum_{(u,i) \in \mathcal{D}_u} -\log(1 - \pi(x_{u,i}, g_{u,i})). \tag{5}$$

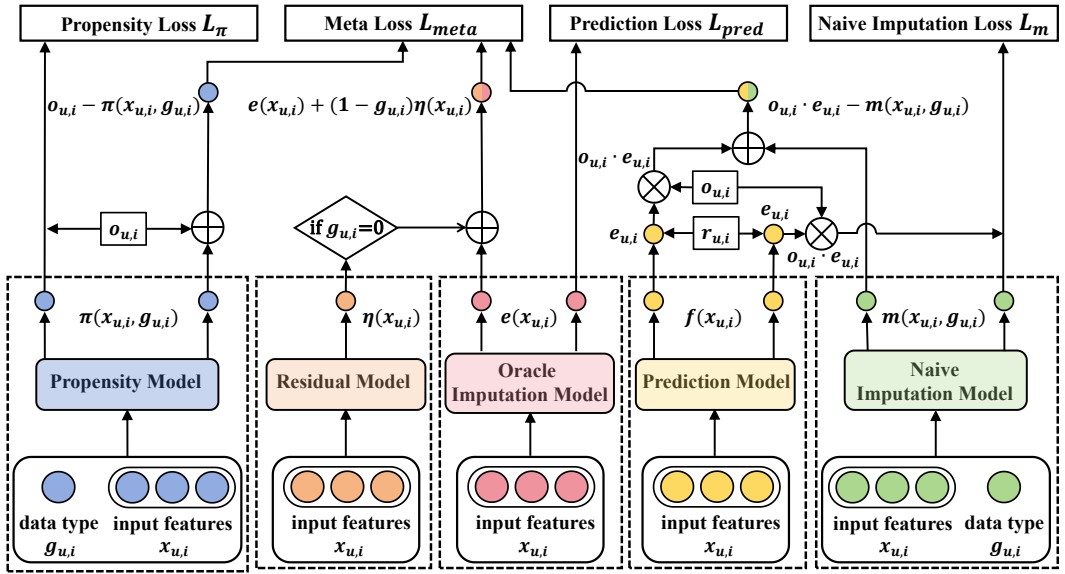

Figure 2: Architecture of MetaDebias to address selection bias in the presence of hidden confounding.

According to the definition of $m(x, g)$ in Equation 2, we adopt the square loss to train the naive imputation model $m(x, g; \phi_m)$ with parameters $\phi_m$, denoted as $\mathcal{L}_m(\phi_m, \theta)$ shown in Equation 6:

$$
\begin{aligned}
\mathcal{L}_m(\phi_m, \theta) &= \frac{1}{|\mathcal{D}|} \sum_{(u,i) \in \mathcal{D}} \left( m\left(x_{u,i}, g_{u,i}\right) - o_{u,i} \cdot e_{u,i} \right)^2 \\
&= \frac{1}{|\mathcal{D}|} \sum_{(u,i) \in \mathcal{D}_0 \cup \mathcal{D}_1} \left( m\left(x_{u,i}, g_{u,i}\right) - e_{u,i} \right)^2 + \frac{1}{|\mathcal{D}|} \sum_{(u,i) \in \mathcal{D}_u} \left( m\left(x_{u,i}, g_{u,i}\right) \right)^2, \quad (6)
\end{aligned}
$$

where $e_{u,i} = L(f(x_{u,i}; \theta), r_{u,i})$ is the prediction error and only available when $o_{u,i} = 1$.

Next, based on Lemma 2, we employ the square loss to train the oracle imputation model $e(x; \phi_e)$ with parameters $\phi_e$ and residual model $\eta(x; \phi_\eta)$ with parameters $\phi_\eta$ given learned propensity model $\phi_\pi$ and imputation model $\phi_m$, denoted as $\mathcal{L}_{meta}(\phi_e, \phi_\eta, \phi_m, \phi_\pi, \theta)$ shown in Equation 7 below:

$$
\begin{aligned}
\mathcal{L}_{meta}(\phi_e, \phi_\eta, \phi_m, \phi_\pi, \theta) = \frac{1}{|\mathcal{D}|} \sum_{(u,i) \in \mathcal{D}} \{ & o_{u,i} \cdot e_{u,i} - m\left(x_{u,i}, g_{u,i}\right) \\
& - [e(x_{u,i}) + (1 - g_{u,i}) \cdot \eta(x_{u,i})] \cdot [o_{u,i} - \pi\left(x_{u,i}, g_{u,i}\right)]\}^2.
\end{aligned} \quad (7)
$$

Furthermore, we adopt the oracle imputation model $e(x; \phi_e)$ to generate prediction errors on the target population $\mathcal{D}$ as the training objective for prediction model $f(x_{u,i}; \theta)$ with parameters $\theta$. The training objective $\mathcal{L}_{pred}(\theta, \phi_e)$ is demonstrated in the following Equation 8:

$$
\mathcal{L}_{pred}(\theta, \phi_e) = \frac{1}{|\mathcal{D}|} \sum_{(u,i) \in \mathcal{D}} e(x_{u,i}; \phi_e \mid \theta), \quad (8)
$$

where $e(x_{u,i}; \phi_e \mid \theta)$ is the learned oracle imputation model training from $\mathcal{L}_{meta}$ loss in Equation 7.

### 3.4 Learning Algorithm

Following previous works [3, 56], we propose a bi-level optimization based learning algorithm for model training. Specifically, we first train the propensity model $\pi(x, g)$ independently by minimizing the $\mathcal{L}_\pi$ loss shown in Equation 5, as its training objective is orthogonal to those of other models. Next, we use bi-level optimization to update the remaining models.

In the bi-level optimization framework, we first assumed update of the naive imputation model $m(x, g; \phi_m)$ and the oracle imputation model $e(x; \phi_e)$, to ensure that the gradient of $\phi'_e(\phi'_m, \theta)$ can

---

**Algorithm 1:** The Proposed MetaDebias Learning Algorithm

---

**Input:** observed ratings $\mathbf{R}^o$.

1   **Pretrain** propensity model $\pi(x, g, \phi_\pi)$

2   **while** *stopping criteria is not satisfied* **do**

3     **for** *number of training steps* **do**

4       Sample a batch of user-item pairs $\{(u_i, i_i)\}_{i=1}^{B_1}$ from $\mathcal{D}$;

5       **Assumed update** naive imputation model: $\phi'_m(\theta) = \phi_m - \alpha_m \nabla_{\phi_m} \mathcal{L}_m \left( m(\phi_m) \mid \theta \right)$ ;

6       **Assumed update** oracle imputation model:
         $\phi'_e(\phi'_m, \theta) = \phi_e - \alpha_e \nabla_{\phi_e} \mathcal{L}_{meta} \left( e(\phi_e) \mid \phi'_m, \theta \right)$;

7       **Update** prediction model: $\theta \leftarrow \theta - \alpha \nabla_\theta \mathcal{L}_{pred} \left( f_{\phi'_e(\phi'_m, \theta)} \right)$ ;

8     **end**

9     **for** *number of training steps* **do**

10       Sample a batch of user-item pairs $\{(u_i, i_i)\}_{i=1}^{B_2}$ from $\mathcal{D}$;

11       **Update** naive imputation model: $\phi_m \leftarrow \phi_m - \alpha_m \nabla_{\phi_m} \mathcal{L}_m \left( m(\phi_m) \mid \theta \right)$ ;

12     **end**

13     **for** *number of training steps* **do**

14       Sample a batch of user-item pairs $\{(u_i, i_i)\}_{i=1}^{B_3}$ from $\mathcal{D}$;

15       **Update** oracle imputation model: $\phi_e \leftarrow \phi_e - \alpha_e \nabla_{\phi_e} \mathcal{L}_{meta} \left( e(\phi_e) \mid \phi_m, \theta \right)$ ;

16       **Update** residual model: $\phi_\eta \leftarrow \phi_\eta - \alpha_\eta \nabla_{\phi_\eta} \mathcal{L}_{meta} \left( \eta(\phi_\eta) \mid \phi_m, \theta \right)$ ;

17     **end**

18 **end**

    **Output:** $\theta$

---

be back propagated through the chain rule to update the prediction model parameter $\theta$. Subsequently, the prediction model $f(x_{u,i}; \theta)$ is updated by minimizing $\mathcal{L}_{pred}(\theta, \phi_e)$ defined in Equation 8.

After updating the prediction model, we adopt joint learning to update the naive imputation model $m(x, g; \phi_m)$ by minimizing $\mathcal{L}_m$ in Equation 6. Simultaneously, the oracle imputation model $e(x; \phi_e)$ and residual model $\eta(x; \phi_\eta)$ are updated by minimizing $\mathcal{L}_{mata}$ in Equation 7. The whole procedure of the learning algorithm is outlined in Algorithm 1, where $\alpha$ represents the learning rate and we use distinct subscripts to correspond to different models.

## 4 Experiments

### 4.1 Experimental Setup

**Dataset and Experimental Details.** Following previous studies [3, 43, 55, 56], we conduct extensive experiments on three public datasets, COAT[2], YAHOO! R3[3], and KUAIREC[4] [11]. COAT contains 6,960 biased ratings and 4,640 unbiased ratings derived from 290 users evaluating 300 items. YAHOO! R3 contains 311,704 biased ratings and 54,000 unbiased ratings derived from 15,400 users evaluating 1,000 items. Both datasets employ a five-point rating scale, and we binarize the ratings greater than 3 as 1 and others as 0. KUAIREC contains 4,676,570 video watching ratios derived from 1,411 users evaluating 3,327 videos. The ratios that greater than 2 are binarized as 1, otherwise as 0. We adopt feature masking to simulate $g_{u,i}$ which measures feature completeness, and $g_{u,i} = 1$ only when the features of both users and items are fully preserved, otherwise $g_{u,i} = 0$, and see Appendix A.3 for more details. Following previous works [3, 28, 29, 34], we randomly split 5% unbiased data from the test set as validation set, and for all methods requiring RCT data, we employ observational data without hidden confounding to pretend RCT data. For evaluation, we employ three widely adopted metrics AUC, Recall@$K$, and NDCG@$K$ to measure debiasing performance, where we set $K = 5$ on COAT and YAHOO! R3, and $K = 50$ on KUAIREC. All the methods are implemented on PyTorch with Adam as the optimizer and NVIDIA A40 as the computing resource, and we tune learning rate in $\{0.0001, 0.0005, 0.001, 0.005, 0.01, 0.05\}$ and weight decay in $[1e - 7, 10]$.

---

[2] https://www.cs.cornell.edu/~schnabts/mnar/

[3] https://webscope.sandbox.yahoo.com

[4] https://github.com/chongminggao/KuaiRec

Table 1: Performance on AUC, Recall@$K$ and NDCG@$K$ on the COAT, YAHOO! R3 and KUAIREC datasets. The best result is bolded and the best results of both types of baseline methods are underlined, where * means statistically significant results (p-value $\leq 0.05$) using the paired-t-test.

| Method | COAT AUC | Recall@5 | NDCG@5 | YAHOO! R3 AUC | Recall@5 | NDCG@5 | KUAIREC AUC | Recall@50 | NDCG@50 |
|---|---|---|---|---|---|---|---|---|---|
| Naive | $0.698_{\pm0.009}$ | $0.478_{\pm0.018}$ | $0.444_{\pm0.014}$ | $0.705_{\pm0.007}$ | $0.638_{\pm0.011}$ | $0.489_{\pm0.009}$ | $0.817_{\pm0.005}$ | $0.830_{\pm0.006}$ | $0.540_{\pm0.009}$ |
| IPS | $0.717_{\pm0.016}$ | $0.483_{\pm0.021}$ | $0.446_{\pm0.020}$ | $0.699_{\pm0.016}$ | $0.642_{\pm0.018}$ | $0.492_{\pm0.016}$ | $0.814_{\pm0.007}$ | $0.834_{\pm0.008}$ | $0.548_{\pm0.015}$ |
| DR | $0.725_{\pm0.012}$ | $0.485_{\pm0.018}$ | $0.448_{\pm0.008}$ | $0.709_{\pm0.014}$ | $0.643_{\pm0.012}$ | $0.498_{\pm0.016}$ | $0.824_{\pm0.005}$ | $0.836_{\pm0.010}$ | $0.550_{\pm0.016}$ |
| Stable-DR | $0.734_{\pm0.009}$ | $0.486_{\pm0.011}$ | $0.452_{\pm0.010}$ | $0.715_{\pm0.013}$ | $0.656_{\pm0.013}$ | $0.515_{\pm0.015}$ | $0.826_{\pm0.007}$ | $0.839_{\pm0.008}$ | $0.560_{\pm0.015}$ |
| TDR | $0.736_{\pm0.008}$ | $0.492_{\pm0.015}$ | $0.458_{\pm0.011}$ | $\underline{0.717}_{\pm0.006}$ | $0.669_{\pm0.015}$ | $\underline{0.525}_{\pm0.014}$ | $0.830_{\pm0.005}$ | $0.846_{\pm0.010}$ | $0.570_{\pm0.015}$ |
| ESMM | $0.690_{\pm0.014}$ | $0.480_{\pm0.023}$ | $0.458_{\pm0.023}$ | $0.710_{\pm0.010}$ | $0.657_{\pm0.016}$ | $0.513_{\pm0.014}$ | $0.826_{\pm0.004}$ | $0.838_{\pm0.005}$ | $0.560_{\pm0.008}$ |
| Multi-IPS | $0.736_{\pm0.008}$ | $0.482_{\pm0.010}$ | $0.456_{\pm0.011}$ | $0.709_{\pm0.007}$ | $0.647_{\pm0.017}$ | $0.502_{\pm0.016}$ | $0.820_{\pm0.004}$ | $0.830_{\pm0.006}$ | $0.548_{\pm0.010}$ |
| Multi-DR | $0.732_{\pm0.008}$ | $0.485_{\pm0.010}$ | $0.447_{\pm0.008}$ | $0.706_{\pm0.009}$ | $0.649_{\pm0.014}$ | $0.506_{\pm0.014}$ | $0.832_{\pm0.003}$ | $0.841_{\pm0.004}$ | $0.562_{\pm0.008}$ |
| ESCM²-IPS | $0.730_{\pm0.009}$ | $0.484_{\pm0.014}$ | $0.451_{\pm0.010}$ | $0.713_{\pm0.014}$ | $0.666_{\pm0.008}$ | $0.520_{\pm0.010}$ | $0.825_{\pm0.004}$ | $0.844_{\pm0.005}$ | $0.567_{\pm0.008}$ |
| ESCM²-DR | $0.737_{\pm0.009}$ | $0.492_{\pm0.011}$ | $0.458_{\pm0.006}$ | $0.715_{\pm0.006}$ | $\underline{0.670}_{\pm0.008}$ | $0.521_{\pm0.008}$ | $0.832_{\pm0.002}$ | $0.841_{\pm0.002}$ | $0.570_{\pm0.003}$ |
| BRD-IPS | $0.733_{\pm0.011}$ | $0.490_{\pm0.013}$ | $0.462_{\pm0.014}$ | $0.712_{\pm0.008}$ | $0.659_{\pm0.016}$ | $0.515_{\pm0.016}$ | $0.833_{\pm0.003}$ | $0.846_{\pm0.004}$ | $0.566_{\pm0.005}$ |
| BRD-DR | $\underline{0.739}_{\pm0.007}$ | $\underline{0.494}_{\pm0.015}$ | $\underline{0.464}_{\pm0.012}$ | $0.714_{\pm0.009}$ | $0.663_{\pm0.013}$ | $0.516_{\pm0.013}$ | $\underline{0.834}_{\pm0.002}$ | $\underline{0.848}_{\pm0.004}$ | $\underline{0.572}_{\pm0.004}$ |
| KD-Label | $0.735_{\pm0.006}$ | $0.488_{\pm0.010}$ | $0.461_{\pm0.010}$ | $0.712_{\pm0.004}$ | $0.664_{\pm0.010}$ | $0.517_{\pm0.006}$ | $0.831_{\pm0.003}$ | $0.841_{\pm0.005}$ | $0.566_{\pm0.008}$ |
| AutoDebias | $0.736_{\pm0.010}$ | $\underline{0.501}_{\pm0.012}$ | $0.465_{\pm0.006}$ | $0.710_{\pm0.008}$ | $0.667_{\pm0.015}$ | $0.520_{\pm0.013}$ | $0.833_{\pm0.003}$ | $0.843_{\pm0.008}$ | $0.558_{\pm0.006}$ |
| LTD-IPS | $0.732_{\pm0.008}$ | $0.483_{\pm0.013}$ | $0.458_{\pm0.011}$ | $0.708_{\pm0.007}$ | $0.660_{\pm0.014}$ | $0.514_{\pm0.013}$ | $0.834_{\pm0.003}$ | $0.847_{\pm0.003}$ | $0.572_{\pm0.004}$ |
| LTD-DR | $0.734_{\pm0.009}$ | $0.485_{\pm0.014}$ | $0.460_{\pm0.010}$ | $0.711_{\pm0.005}$ | $0.662_{\pm0.016}$ | $0.516_{\pm0.015}$ | $0.835_{\pm0.002}$ | $0.848_{\pm0.003}$ | $0.570_{\pm0.003}$ |
| Bal-IPS | $0.733_{\pm0.010}$ | $0.486_{\pm0.013}$ | $0.462_{\pm0.011}$ | $0.708_{\pm0.012}$ | $0.665_{\pm0.016}$ | $0.515_{\pm0.012}$ | $0.833_{\pm0.003}$ | $0.846_{\pm0.005}$ | $0.573_{\pm0.005}$ |
| Bal-DR | $0.735_{\pm0.011}$ | $0.490_{\pm0.011}$ | $0.464_{\pm0.010}$ | $0.708_{\pm0.014}$ | $0.668_{\pm0.014}$ | $0.517_{\pm0.014}$ | $0.834_{\pm0.004}$ | $0.847_{\pm0.006}$ | $0.568_{\pm0.006}$ |
| Res-IPS | $0.738_{\pm0.010}$ | $0.494_{\pm0.012}$ | $0.465_{\pm0.010}$ | $0.718_{\pm0.004}$ | $0.675_{\pm0.008}$ | $0.534_{\pm0.005}$ | $0.836_{\pm0.002}$ | $0.850_{\pm0.005}$ | $0.578_{\pm0.002}$ |
| Res-DR | $\underline{0.740}_{\pm0.006}$ | $0.498_{\pm0.010}$ | $\underline{0.467}_{\pm0.009}$ | $\underline{0.720}_{\pm0.004}$ | $\underline{0.678}_{\pm0.006}$ | $\underline{0.538}_{\pm0.006}$ | $\underline{0.838}_{\pm0.002}$ | $0.852_{\pm0.004}$ | $\underline{0.580}_{\pm0.003}$ |
| MetaDebias | $\mathbf{0.746}^{*}_{\pm0.007}$ | $\mathbf{0.510}^{*}_{\pm0.008}$ | $\mathbf{0.473}^{*}_{\pm0.010}$ | $\mathbf{0.722}^{*}_{\pm0.004}$ | $\mathbf{0.688}^{*}_{\pm0.005}$ | $\mathbf{0.544}^{*}_{\pm0.005}$ | $\mathbf{0.840}^{*}_{\pm0.001}$ | $\mathbf{0.857}^{*}_{\pm0.004}$ | $\mathbf{0.584}^{*}_{\pm0.003}$ |

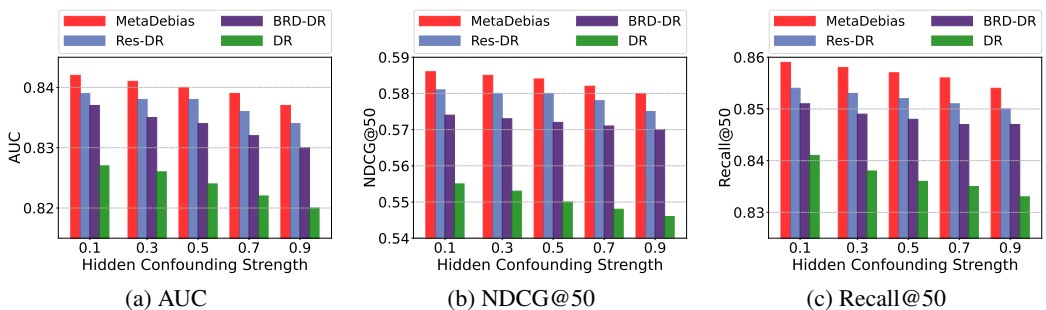

(a) AUC    (b) NDCG@50    (c) Recall@50

Figure 3: Effects of hidden confounding strength on the KUAIREC dataset.

**Baselines.** Two-layer multi-layer perceptron are used as the base model, and we compare proposed methods with both RCT data-free and RCT data-based methods.

• **RCT data-free Methods**: Without RCT data, the baselines includes IPS-based methods: **IPS** [46], **ESMM** [37, 57], **Multi-IPS** [64], **ESCM²-IPS** [52], **BRD-IPS** [10], and DR-based methods: **DR** [55], **Stable-DR** [31], **TDR** [27], **Multi-DR** [64], **ESCM²-DR** [52], , **BRD-DR** [10].

• **RCT data-based Methods**: Based on RCT data, the baselines include model selection based methods: **KD-Label** [34], **AutoDebias** [3], **LTD-IPS** [56],**LTD-DR** [56], and model calibration based methods: **Bal-IPS** [29], **Bal-DR** [29], **Res-IPS** [28], **Res-DR** [28].

## 4.2 Experimental Results

**Performance Comparison.** We compare our proposed MetaDebias with existing methods and the results are shown in Table 1. First, all debiasing models outperform the base model which adopts naive empirical risk minimization, indicating the necessity for debiasing. Second, BRD-DR and Res-DR which are two representative methods for eliminating hidden confounding serve as the most competitive baselines, implying that ignoring hidden confounding leads to inevitable performance degradation. Third, compared to RCT data-free methods, RCT data-based methods do not demonstrate significant improvements across the three datasets, indicating that effective model selection or model calibration cannot be achieved using MNAR data without hidden confounding. Furthermore, MetaDebias exhibits significantly superior overall performance on all three datasets, which validates that our method achieves state-of-the-art performance in the presence of hidden confounding leveraging heterogeneous observational data without the utilization of RCT data.

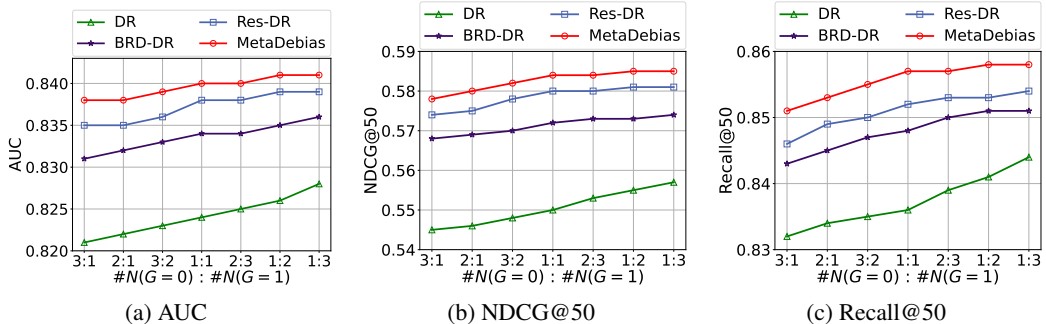

Figure 4: Effects of varying proportions of heterogeneous observational data on the KUAIREC dataset.

Table 2: Effects of training dataset size on NDCG@$K$ on the KUAIREC and YAHOO! R3 datasets.

| NDCG@50 | Training Set Size (%) | | | | | |
|---|---|---|---|---|---|---|
| Method | 10 | 20 | 40 | 60 | 80 | 100 |
| DR | 0.532 | 0.534 | 0.538 | 0.542 | 0.548 | 0.550 |
| ESCM$^2$-DR | 0.556 | 0.560 | 0.563 | 0.565 | 0.568 | 0.570 |
| BRD-DR | 0.567 | 0.568 | 0.570 | 0.570 | 0.571 | 0.572 |
| Bal-DR | 0.552 | 0.554 | 0.558 | 0.562 | 0.565 | 0.568 |
| Res-DR | 0.569 | 0.571 | 0.572 | 0.574 | 0.577 | 0.580 |
| MetaDebias | 0.576 | 0.577 | 0.578 | 0.579 | 0.581 | 0.584 |

| NDCG@5 | Training Set Size (%) | | | | | |
|---|---|---|---|---|---|---|
| Method | 10 | 20 | 40 | 60 | 80 | 100 |
| DR | 0.490 | 0.492 | 0.494 | 0.494 | 0.496 | 0.498 |
| ESCM$^2$-DR | 0.506 | 0.510 | 0.513 | 0.517 | 0.519 | 0.521 |
| BRD-DR | 0.500 | 0.503 | 0.506 | 0.509 | 0.511 | 0.516 |
| Bal-DR | 0.502 | 0.505 | 0.506 | 0.509 | 0.513 | 0.517 |
| Res-DR | 0.523 | 0.527 | 0.528 | 0.531 | 0.533 | 0.538 |
| MetaDebias | 0.534 | 0.536 | 0.538 | 0.540 | 0.541 | 0.544 |

(a) NDCG@50 on the KUAIREC dataset

(b) NDCG@5 on the YAHOO! R3 dataset

**In-depth Analysis.** We conduct in-depth analysis to further explore the effect of hidden confounding strength, the proportions of heterogeneous observational data, and training data size on performance. Moreover, we further explore the performance under conditions where RCT data is available, which is consistent with the problem setup in prior works. See Appendix A.4 for more experimental results.

**The Hidden Confounding Strength.** Figure 3 shows the impact of hidden confounding strength, where we simulate higher hidden confounding strength by masking more features. We observe all methods exhibit performance degradation as the hidden confounding gets stronger, with vanilla DR exhibiting the poorest performance, highlighting the necessity of removing hidden confounding. The performance of BRD-DR deteriorates significantly, primarily due to the violation of imposed strong model assumptions, while Res-DR demonstrates competitive performance. In addition, the proposed MetaDebias stably outperforms the baselines across varying hidden confounding strength, implying our method is able to effectively address strong hidden confounding using only observational data.

**The Proportion of Heterogeneous Observational Data.** We explore the impact of varying proportions of heterogeneous observational data on the KUAIREC dataset in Figure 4, where $\#N(G = 0)$ and $\#N(G = 1)$ denote the quantity of training data with and without hidden confounding. First, we observe the performance of DR varies significantly under different proportions, indicating the pronounced disparities between the heterogeneous observational data. Besides, performances of all methods are enhanced when the proportion of data without hidden confounding increases, this is because more feature information aids in accurate propensity and imputation model learning. Furthermore, MetaDebias demonstrates the best performance, indicating our method can achieve effective debiasing performance across various observational data proportions, and the potential for application in a range of real-world scenarios.

**The Training Dataset Size.** Table 2 explores the impact of training set size on NDCG@$K$ on both KUAIREC and YAHOO! R3 datasets. We find that the performance of all methods declines as the size of training set decreases, highlighting the importance of training data size on model training. Besides, the proposed MetaDebias consistently outperforms the baselines across varying training data size especially when the size is extremely small such as only 10%, which indicates our method that explicitly models oracle prediction errors on the entire space fully exploits the heterogeneous observational data and is still effective even with small training set size.

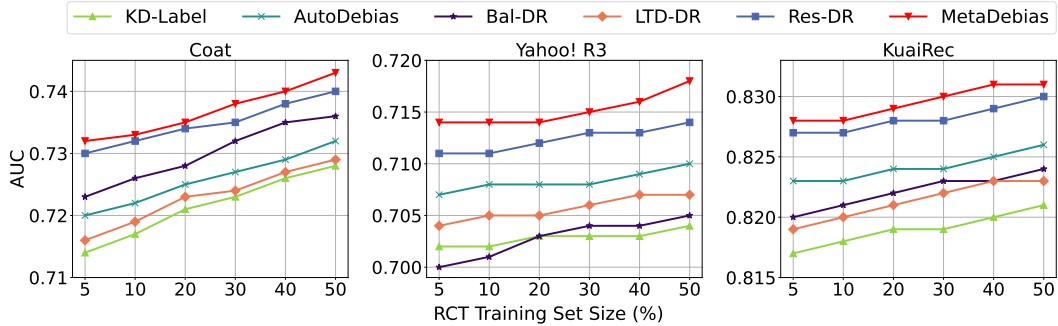

Figure 5: Effects of varying RCT training set size on AUC on three benchmark datasets.

**The Impact of RCT Training Data.** In Figure 5, we explore the debiasing performance when RCT data is available, which is consistent with the problem setup in prior works [28, 29]. Specifically, the training dataset is composed of two parts: an observational dataset with hidden confounding and a small RCT dataset. For implementing our proposed MetaDebias, we label the RCT sample as $g_{u,i} = 1$, as the RCT data can be regarded a special case of MNAR data without hidden confounding. We find that the performances of all methods exhibit an increasing trend as the size of RCT training set increases, which is consistent with previous research findings. Moreover, the proposed MetaDebias stably outperforms all the baseline methods with varying RCT training set size, which indicates our method remains superior to the existing approaches under the previous problem setup, further validating the effectiveness of our method.

## 5 Conclusion

In this study, we investigate the problem of selection bias in the presence of hidden confounding. First, we argue that existing methods are challenging to be applied in real-world scenarios, as they either rely on strong assumptions on hidden confounding strength or depend on the costly RCT data. To tackle this issue, we propose to adopt heterogeneous observational datasets which are more likely to be collected to address hidden confounding and claim that such setup is more aligned with practical scenarios. Then, we propose a meta-learning based debiasing method called MetaDebias, which explicitly models the oracle error imputation and additional bias induced by hidden confounders, and we adopt bi-level optimization with assumed update for model training. Extensive experiments conducted on three public datasets validate our method achieves state-of-the-art performance, regardless of the availability of RCT data. A limitation of this work lies in the assumption that all the confounders can be included through sufficient feature collection. Even though it is possible to collect hundreds of features in industrial scenarios, and some features like consumption records may potentially represent hard-to-obtain features like salary, it is still challenging to guarantee that all confounders for partial samples have been included. In future work, we will explore how to further relax this theoretically feasible assumption.

## Acknowledgments and Disclosure of Funding

The authors thank the anonymous reviewers for their valuable comments. This work was supported by the National Natural Science Foundation of China under Grant 623B2002, Grant 62106266 and Grant U22B2048.

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

# A    Appendix / supplemental material

## A.1    Related Work

**Debiased Recommendation.** Recommender systems play an important role in mitigating information overload which are trained on users' historical feedback, but selection bias inherent in the collected data impede the algorithms from accurately capturing users' true preferences [17, 53, 67]. If we naively adopt empirical risk minimization to train prediction model without debiasing, it will achieve sub-optimal prediction performance [46, 55, 60]. To address the selection bias and achieve unbiased learning, the error imputation based (EIB) method [38, 48] treats selection bias as a missing data problem, imputes the missing data, and subsequently trains prediction model combining with the imputed data. The inverse propensity scoring (IPS) method [36, 45, 46] estimates the probability of an user-item pair being observed which is called propensity and performs inverse propensity weighting for the observed samples. The doubly robust (DR) method [39, 43, 55] combines both error imputation and inverse propensity weighting, achieving superior performance while exhibiting enhanced robustness. Owing to the extensive adoption of IPS and DR, numerous variants have been proposed [26, 31, 49]. Specifically, some approaches focus on bias and variance trade-off [7, 12, 27], while others integrate with multi-task learning to address data sparsity [52, 64]. Additionally, some works explore how to learn more appropriate propensity scores [3, 30, 32, 56]. However, the aforementioned methods fail to achieve unbiased when some features that simultaneous affect both user selection mechanism and the feedback remain unobserved, which is called hidden confounding. Despite recent works proposed to address observed confounding effects [14, 47], tackling hidden confounding remains a challenging task. To fill this gap, sensitivity analysis based methods are proposed, where worst case optimization is used to mitigate the hidden confounding [10, 66]. However, the validity of such methods relies on the assumption on hidden confounding effect that true propensities are near and can be bounded by the estimated values, and such approaches become ineffective when this assumption is violated. Motivated by this, recent works propose to leverage a small amount of RCT data for model calibration, where the calibrated estimator serves an unbiased estimator of the ideal loss [28, 29]. Observing that the acquisition cost of RCT data is prohibitively high, it is equally challenging to implement such calibration based methods in real-world applications. In this paper, we propose a more practical solution to address hidden confounding using heterogeneous observational data for explicitly modeling the oracle prediction errors.

**Causal Inference under Hidden Confounding.** Hidden confounding is a widely studied topic in causal inference literature [1, 50, 65], and there are two primary approaches to addressing this issue. One approach involves leveraging additional information to achieve unbiased estimation of the target causal quantity, for example the instrumental variables [15], front door adjustment [40], negative control [33] and data fusion [24, 58]. The second approach based on sensitive analysis, on the other hand, aims to estimate appropriate bounds for the target causal effects, rather than pursuing precise point estimation [6, 23, 41, 51]. However, in practical applications, it is hard to identify suitable instruments or mediators that meet the required criteria [15, 20], while the sensitive analysis based method heavily relies on strong assumption of the hidden confounding effect, which is also hard to satisfy. The data fusion approach aims to combine data from observational studies and randomized trials, and can be broadly classified into three categories: statistical test based, correction based, and weighted based. Statistical test based methods introduce statistical tests to compare the causal effects estimated from observational studies and randomized trials, thereby detecting and mitigating hidden confounding [8, 9, 18, 19]. Correction based methods are proposed to correct the biased causal effect estimation derived from observational data using the unbiased RCT data [13, 24, 63], and an efficient integrative estimator is established based on semi-parametric theory [62], and further integrated with machine learning models [58]. Weighted methods propose to train a biased estimator using observational data, train an unbiased estimator using RCT data, and take the weighted average of these two estimators as the final result [5, 42, 61]. A limitation of data fusion methods is the availability of RCT data, as the cost of obtaining RCT data is prohibitively high. Moreover, for the correction based method, the randomized trial and observational study should share the same support sets. When the support sets differ and only a partial overlap exists, additional strong parametric assumptions are required for extrapolation, for instance, the hidden confounding effect is assumed to be a linear function [24]. In contrast to the aforementioned methods, our proposed approach is more readily feasible in real-world scenarios, as it does not impose assumption on hidden confounding effect and the required heterogeneous observational data is relatively more accessible.

## A.2 Proofs

**Lemma 1** *The relationship between the proposed propensity score $\pi(x, g)$, naive error imputation $m(x, g)$ and the oracle error imputation $\mathbb{E}\left[e_{u,i} \mid x_{u,i}\right]$ is as followed:*

$$m(x, g) = \{\mathbb{E}\left[e_{u,i} \mid x_{u,i} = x\right] + (1 - g)\eta(x)\} \cdot \pi(x, g),$$

*where $\eta(x) = \mathbb{E}\left[e_{u,i} \mid x_{u,i} = x, g_{u,i} = 0, o_{u,i} = 1\right] - \mathbb{E}\left[e_{u,i} \mid x_{u,i} = x\right]$.*

*Proof.* Recall that:

$$m(x, g) = \mathbb{E}[o_{u,i} \cdot e_{u,i} \mid x_{u,i} = x, g_{u,i} = g],$$
$$\pi(x, g) = \mathbb{P}\left(o_{u,i} = 1 \mid x_{u,i} = x, g_{u,i} = g\right).$$

Then, when $g$ equals 1, we have:

$$
\begin{aligned}
m(x, g = 1) &= \mathbb{E}\left[o_{u,i} \cdot e_{u,i} \mid x_{u,i} = x, g_{u.i} = 1\right] \\
&= \mathbb{E}\left[e_{u,i} \mid x_{u,i} = x, g_{u.i} = 1, o_{u,i} = 1\right] \cdot \mathbb{P}\left(o_{u,i} = 1 \mid x_{u,i} = x, g_{u.i} = 1\right) \\
&= \mathbb{E}\left[e_{u,i} \mid x_{u,i} = x, g_{u.i} = 1\right] \cdot \mathbb{P}\left(o_{u,i} = 1 \mid x_{u,i} = x, g_{u.i} = 1\right) \\
&= \mathbb{E}\left[e_{u,i} \mid x_{u,i} = x\right] \cdot \mathbb{P}\left(o_{u,i} = 1 \mid x_{u,i} = x, g_{u.i} = 1\right) \\
&= \mathbb{E}\left[e_{u,i} \mid x_{u,i} = x\right] \cdot \pi(x, g = 1).
\end{aligned}
$$

The first equation is the definition of $m(x, g)$, the second equation is the law of total probability, and the third equation holds as the conditional independence $e_{u,i} \perp\!\!\!\perp o_{u,i} \mid x_{u,i}, g_{u,i} = 1$. The fourth equation holds because $e_{u,i} = L(f(x_{u,i}), r_{u,i})$ is independent of the data type variable $g_{u,i}$, and the last equation is the definition of $\pi(x, g)$.

Similarly, when $g$ equals 0, the equivalent equation holds:

$$
\begin{aligned}
m(x, g = 0) &= \mathbb{E}\left[o_{u,i} \cdot e_{u,i} \mid x_{u,i} = x, g_{u.i} = 0\right] \\
&= \mathbb{E}\left[e_{u,i} \mid x_{u,i} = x, g_{u.i} = 0, o_{u,i} = 1\right] \cdot \mathbb{P}\left(o_{u,i} = 1 \mid x_{u,i} = x, g_{u.i} = 0\right) \\
&= \mathbb{E}\left[e_{u,i} \mid x_{u,i} = x, g_{u.i} = 0, o_{u,i} = 1\right] \cdot \pi(x, g = 0) \\
&= \{\mathbb{E}\left[e_{u,i} \mid x_{u,i} = x\right] + \eta(x)\} \cdot \pi(x, g = 0).
\end{aligned}
$$

The second equation is the law of total probability, the third equation is the definition of $\pi(x, g)$, and the last equation is the definition of $\eta(x)$ which is equivalently expressed as:

$$\mathbb{E}\left[e_{u,i} \mid x_{u,i} = x, g_{u,i} = 0, o_{u,i} = 1\right] = \eta(x) + \mathbb{E}\left[e_{u,i} \mid x_{u,i} = x\right].$$

To sum up, the equation holds:

$$m(x, g) = \{\mathbb{E}\left[e_{u,i} \mid x_{u,i} = x\right] + (1 - g)\eta(x)\} \cdot \pi(x, g).$$

$\square$

**Lemma 2** *The relationship between the propensity score $\pi(x, g)$, naive error imputation $m(x, g)$, observation $o_{u,i}$, naive prediction error $o_{u,i} \cdot e_{u,i}$ and oracle error imputation $\mathbb{E}\left[e_{u,i} \mid x_{u,i}\right]$ is:*

$$o_{u,i} \cdot e_{u,i} - m(x, g) = \{\mathbb{E}\left[e_{u,i} \mid x_{u,i} = x\right] + (1 - g)\eta(x)\} \cdot \{o_{u,i} - \pi(x, g)\} + \xi,$$

*where $\xi = o_{u,i} \cdot \{e_{u,i} - \{\mathbb{E}[e_{u,i} \mid x_{u,i} = x] + (1 - g)\eta(x)\}\}$ with $\mathbb{E}[\xi \mid x, g] = 0$.*

*Proof.* We prove this Lemma in two steps. Initially, we establish the equality relation in the Lemma $o_{u,i} \cdot e_{u,i} - m(x, g) = \{\mathbb{E}\left[e_{u,i} \mid x_{u,i} = x\right] + (1 - g)\eta(x)\} \cdot \{o_{u,i} - \pi(x, g)\} + \xi$. Subsequently, we prove that the conditional expectation of the noise term is zero, i.e., $\mathbb{E}[\xi \mid x, g] = 0$.

To start with, based on **Lemma 1**, the equation holds:

$$m(x, g) = \{\mathbb{E}\left[e_{u,i} \mid x_{u,i} = x\right] + (1 - g)\eta(x)\} \cdot \pi(x, g).$$

According to the definition of $\xi$, we have:

$$o_{u,i} \cdot e_{u,i} = \xi + o_{u,i} \cdot \{\{\mathbb{E}[e_{u,i} \mid x_{u,i} = x] + (1 - g)\eta(x)\}\}.$$

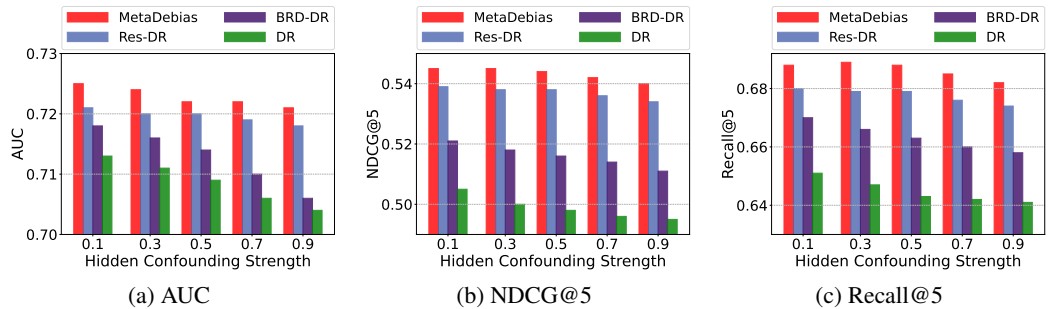

Figure 6: Effects of hidden confounding strength on the YAHOO! R3 dataset.

The subtraction of the above two equations yields the following:

$$o_{u,i} \cdot e_{u,i} - m(x,g) = \{\mathbb{E}\left[e_{u,i} \mid x_{u,i} = x\right] + (1-g)\eta(x)\} \cdot \{o_{u,i} - \pi(x,g)\} + \xi.$$

Next we demonstrate that $\xi$ possesses the zero mean property. Recall that:

$$\begin{aligned}
\xi &= o_{u,i} \cdot \{e_{u,i} - \{\mathbb{E}[e_{u,i} \mid x_{u,i} = x] + (1-g)\eta(x)\}\} \\
&= o_{u,i} \cdot e_{u,i} - o_{u,i} \cdot \{\{\mathbb{E}[e_{u,i} \mid x_{u,i} = x] + (1-g)\eta(x)\}\}.
\end{aligned}$$

Then the conditional expectation of $\xi$ is:

$$\begin{aligned}
\mathbb{E}\left[\xi \mid x,g\right] &= \mathbb{E}\left[o_{u,i} \cdot e_{u,i} \mid x,g\right] - \mathbb{E}\left[o_{u,i} \cdot \{(\mathbb{E}\left[e_{u,i} \mid x_{u,i} = x\right] + (1-g)\eta(x))\} \mid x,g\right] \\
&= m(x,g) - \mathbb{E}\left[o_{u,i} \cdot \{\mathbb{E}\left[e_{u,i} \mid x_{u,i} = x\right] + (1-g)\eta(x)\} \mid x,g\right] \\
&= m(x,g) - \{\mathbb{E}\left[e_{u,i} \mid x_{u,i} = x\right] + (1-g)\eta(x)\} \cdot \mathbb{E}\left[o_{u,i} \mid x_{u,i} = x, g_{u,i} = g\right] \\
&= m(x,g) - \{\mathbb{E}\left[e_{u,i} \mid x_{u,i} = x\right] + (1-g)\eta(x)\} \cdot \mathbb{P}\left(o_{u,i} = 1 \mid x_{u,i} = x, g_{u,i} = g\right) \\
&= m(x,g) - \{\mathbb{E}\left[e_{u,i} \mid x_{u,i} = x\right] + (1-g)\eta(x)\} \cdot \pi(x,g) \\
&= 0.
\end{aligned}$$

The first equation is the definition of $\xi$, the second equation is the the definition of $m(x,g)$, and the third equation holds because $\mathbb{E}\left[e_{u,i} \mid x_{u,i} = x\right] + (1-g)\eta(x)$ is a constant given $x$ and $g$. The fourth equation arises from the fact that the treatment variable $o_{u,i}$ is a binary variable taking values of 0 or 1, and the fifth equation is the definition of $\pi(x,g)$. The last equation holds based on **Lemma 1**.

$\square$

### A.3 More Experimental Details

We introduce more experimental details about feature preparations here. In particular, we first employ Matrix Factorization (MF) [25] to transform the ID-based data format into the feature-based format and regard the derived features as complete features. Next, we simulate the hidden confounding effect induced by insufficient feature collection process through feature masking. The condition $g_{u,i} = 1$ is satisfied only when both user and item features are sufficiently preserved, i.e., there are no masked values in their respective feature vectors; otherwise, it equals 0. At the outset, we set the proportion of two types of heterogeneous observational data at 1:1, and the impact of varying proportions is explored in the following in-depth analysis.

### A.4 More Experimental Results

In this subsection, we will present more experimental results that could not be accommodated within the main text due to space limitations. The main text primarily elucidates the results obtained on the KUAIREC dataset. Supplementary to this, the appendix predominantly supplements experimental findings on the YAHOO! R3 dataset, which similarly explores the impact of hidden confounding strength, the proportions of the heterogeneous observational data, and the data sparsity. Apart from these, we conduct an additional investigation into the algorithm runtime on three datasets to explore the computational resource demands of different methods. The findings are as follows.

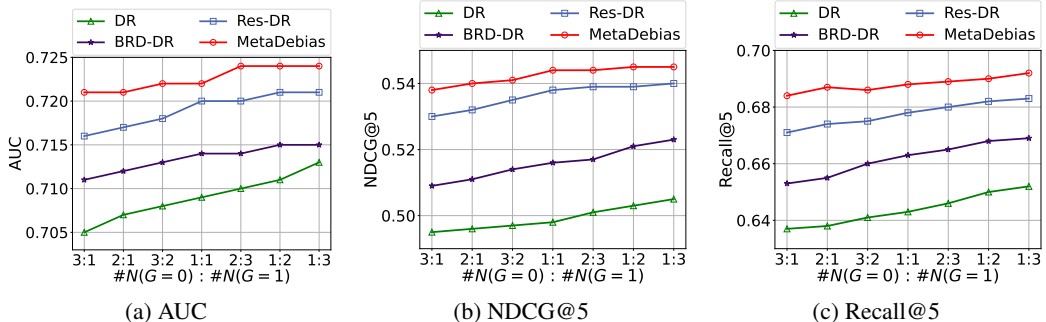

Figure 7: Effects of varying proportions of heterogeneous data on the YAHOO! R3 dataset.

Table 3: Effects of training data size on AUC on the KUAIREC and YAHOO! R3 datasets.

| AUC | Training Set Size (%) | | | | | |
|---|---|---|---|---|---|---|
| Method | 10 | 20 | 40 | 60 | 80 | 100 |
| DR | 0.812 | 0.815 | 0.818 | 0.820 | 0.821 | 0.824 |
| ESCM$^2$-DR | 0.816 | 0.819 | 0.822 | 0.827 | 0.830 | 0.832 |
| BRD-DR | 0.827 | 0.830 | 0.830 | 0.832 | 0.833 | 0.834 |
| Bal-DR | 0.825 | 0.827 | 0.828 | 0.829 | 0.830 | 0.834 |
| Res-DR | 0.830 | 0.832 | 0.834 | 0.836 | 0.837 | 0.838 |
| MetaDebias | 0.836 | 0.837 | 0.837 | 0.838 | 0.839 | 0.840 |

| AUC | Training Set Size (%) | | | | | |
|---|---|---|---|---|---|---|
| Method | 10 | 20 | 40 | 60 | 80 | 100 |
| DR | 0.700 | 0.702 | 0.704 | 0.705 | 0.707 | 0.709 |
| ESCM$^2$-DR | 0.705 | 0.708 | 0.710 | 0.711 | 0.713 | 0.715 |
| BRD-DR | 0.704 | 0.706 | 0.708 | 0.710 | 0.712 | 0.714 |
| Bal-DR | 0.702 | 0.704 | 0.705 | 0.705 | 0.706 | 0.708 |
| Res-DR | 0.712 | 0.714 | 0.714 | 0.716 | 0.718 | 0.720 |
| MetaDebias | 0.720 | 0.721 | 0.722 | 0.721 | 0.722 | 0.722 |

(a) AUC on the KUAIREC dataset

(b) AUC on the YAHOO! R3 dataset

**The Hidden Confounding Strength.** Figure 6 demonstrates the debiasing performance with varying hidden confounding strength on the YAHOO! R3 dataset. Similar to the findings observed in the KUAIREC dataset shown in the main text, on the YAHOO! R3 dataset, an increase in hidden confounding strength leads to a decrease in performance across all methods. Meanwhile, the performance deterioration of the BRD-DR is more pronounced, particularly in terms of AUC metric, attributed to the violation of the required assumptions. Furthermore, the proposed MetaDebias demonstrates superior performance across all three evaluation metrics, and its performance shows minimal degradation as the hidden confounding strength increases. This indicates the superiority of our approach over existing methods, and robustness to the hidden confounding strength.

**The Proportion of Heterogeneous Observational Data.** Figure 7 shows the prediction performance with varying proportions of heterogeneous observational data on the YAHOO! R3 dataset, where varying data proportions simulate potential data collection scenarios under different real-world conditions. From the figure, we observe an upward trend in the performance of all methods with an increase in the proportion of data without hidden confounding, indicating that the selection bias inherent in such observational data without hidden confounding is more amenable to be addressed. Moreover, MetaDebias outperforms other baselines and exhibits insensitivity to data proportions, which underscores that our method is capable of handling hidden confounding in various possible data combinations and can be broadly applied to a wide range of real-world scenarios.

**The Training Dataset Size.** We explore the impact of training set size on prediction performance in Tables 3 and 4, which report the AUC and Recall@$K$ with varying training set size on both KUAIREC and YAHOO! R3 dataset. The presented data represents the mean values obtained from 10 random replicate experiments. We find that MetaDebias demonstrates superior performance across varying training data size, highlighting the efficacy of our method. Meanwhile, MetaDebias exhibits relatively minor fluctuations with changes in dataset volume, particularly evident in the YAHOO! R3 dataset, where the AUC of our method consistently maintains superior performance even amidst substantial variations in training data volume, such as a change from 100% to 10%. This further demonstrates the stability of our approach even with small the training data size.

**The Computational Resource Demands.** We investigate the algorithm training and inference time on three datasets in Table 5. Despite the involvement of five models in the training process, the comparison with other baseline methods reveals that the training time of the proposed approach is

Table 4: Effects of training dataset size on Recall@$K$ on the KUAIREC and YAHOO! R3 datasets.

| Recall@50 | Training Set Size (%) | | | | | |
|---|---|---|---|---|---|---|
| Method | 10 | 20 | 40 | 60 | 80 | 100 |
| DR | 0.824 | 0.826 | 0.828 | 0.829 | 0.833 | 0.836 |
| ESCM$^2$-DR | 0.829 | 0.832 | 0.835 | 0.837 | 0.839 | 0.841 |
| BRD-DR | 0.843 | 0.844 | 0.845 | 0.845 | 0.847 | 0.848 |
| Bal-DR | 0.838 | 0.839 | 0.840 | 0.842 | 0.844 | 0.847 |
| Res-DR | 0.844 | 0.845 | 0.846 | 0.846 | 0.849 | 0.852 |
| MetaDebias | 0.846 | 0.847 | 0.849 | 0.851 | 0.854 | 0.857 |

| Recall@5 | Training Set Size (%) | | | | | |
|---|---|---|---|---|---|---|
| Method | 10 | 20 | 40 | 60 | 80 | 100 |
| DR | 0.634 | 0.635 | 0.636 | 0.637 | 0.640 | 0.643 |
| ESCM$^2$-DR | 0.652 | 0.656 | 0.660 | 0.663 | 0.666 | 0.670 |
| BRD-DR | 0.648 | 0.653 | 0.656 | 0.658 | 0.660 | 0.663 |
| Bal-DR | 0.652 | 0.655 | 0.657 | 0.660 | 0.665 | 0.668 |
| Res-DR | 0.664 | 0.668 | 0.670 | 0.673 | 0.675 | 0.678 |
| MetaDebias | 0.673 | 0.674 | 0.677 | 0.681 | 0.683 | 0.688 |

(a) Recall@50 on the KUAIREC dataset

(b) Recall@5 on the YAHOO! R3 dataset

Table 5: Comparison of training time (minutes) and inference time (milliseconds per sample) on the COAT, YAHOO! R3 and KUAIREC datasets.

| Method | COAT | | YAHOO! R3 | | KUAIREC | |
|---|---|---|---|---|---|---|
| | Training | Inference | Training | Inference | Training | Inference |
| DR | 0.793 | 0.498 | 6.599 | 0.258 | 219.948 | 0.274 |
| TDR | 0.693 | 0.472 | 6.227 | 0.265 | 135.778 | 0.285 |
| Multi-DR | 0.552 | 0.515 | 5.906 | 0.283 | 238.706 | 0.265 |
| ESCM$^2$-DR | 0.462 | 0.578 | 1.098 | 0.262 | 104.668 | 0.782 |
| BRD-DR | 0.682 | 0.347 | 2.973 | 0.261 | 198.872 | 0.249 |
| KD-Label | 1.124 | 0.519 | 2.449 | 0.358 | 163.358 | 0.239 |
| AutoDebias | 1.182 | 0.395 | 1.662 | 0.243 | 204.555 | 0.632 |
| LTD-DR | 2.351 | 0.459 | 1.832 | 0.262 | 253.277 | 0.267 |
| Bal-DR | 2.912 | 0.513 | 1.702 | 0.236 | 157.784 | 0.257 |
| Res-DR | 0.674 | 0.387 | 3.927 | 0.288 | 165.444 | 0.244 |
| MetaDebias | 1.916 | 0.521 | 6.375 | 0.263 | 174.866 | 0.254 |

acceptable, particularly on the large-scale dataset KuaiRec. A potential reason for the relatively long training time required by proposed method is that the bi-level optimization process with assumed updates results in multiple gradient computations throughout the training procedure.

## A.5 Boarder Impacts

Recommender systems (RS) serve as an effective tool for mitigating information overload, yielding significant economic benefits by accurately recommending items of interest to individual user. However, in the data collection process of RS, there inevitably exists a significant amount of bias, rendering prediction models trained on the collected historical feedback incapable of capturing users' true preferences, where selection bias is a particularly common and extensively studied issue. Recently, an increasing number of studies have further focused on and emphasized the impact of hidden confounding, which aims to achieve better prediction performance. In this paper, we propose a more practical solution to effectively address selection bias in the presence of hidden confounding. Specifically, we propose to leverage heterogeneous observational datasets, which is more readily available in real-world scenarios, while not imposing additional assumptions or requiring RCT data. This indicates that our method has the potential to be applied in real-world recommendation scenarios, offering opportunities to enhance economic benefits for businesses.

