# OpenReview forum: "Addressing Hidden Confounding with Heterogeneous Observational Datasets for Recommendation"
_NeurIPS.cc/2024/Conference — NeurIPS 2024 poster_

### Official Review · Reviewer_Yf6Q · 2024-07-09

**Soundness:** 2
**Presentation:** 2
**Contribution:** 2
**Rating:** 7
**Confidence:** 2

**Summary:**

The paper studies the problem of selection bias due to hidden confounding in recommendation systems. Previous methods struggle with real-world application due to reliance on strong assumptions or unavailable RCT data. The proposed solution, MetaDebias, leverages heterogeneous observational data, which is more practical and readily available. The approach involves a meta-learning framework that uses both confounded observational data and unconfounded observational data for model training. Experiments show that MetaDebias consistently outperforms baseline methods across various metrics and conditions.

**Strengths:**

(S1) The paper compares the proposed approach against many existing methods on 3 benchmark recommendation datasets under several commonly used evaluation metrics.

**Weaknesses:**

(W1) I found the use of the term heterogenous to describe the observational data a bit vague and misleading. I think the authors should make very clear from the beginning that they are working with one dataset where they can observe everything (no latents) and one dataset where some variables are unobserved.

(W2) I think the related work section on causal inference (Appendix A.1) could be improved. In particular, it is not clear what is the weakness of the data fusion methods since they are clustered with IVs and negative controls. Moreover, I think some relevant works that use randomized trials to address hidden confounding are missing, for example [1] and [2].

(W3) The (empirical) performance improvement over previous methods is not so strong.

(Minor) Typo in Lemma 1, should be "as follows" not "as followed".

[1] Hidden yet quantifiable: A lower bound for confounding strength using randomized trials. De Bartolomeis et al. AISTATS '24


[2] Falsification before Extrapolation in Causal Effect Estimation. Hussain et al. NeurIPS '22

**Questions:**

(Q1) Could you explain with more details how are you assessing the statistical significance of the performance improvement over the previous methods?

**Limitations:**

Yes

---

> ### Author Rebuttal · Authors · 2024-08-07
>
> We sincerely appreciate the reviewer’s great efforts and insightful comments to improve our manuscript. Below, we hope to address your concerns and questions to improve the clarity and readability of our paper.
>
> > **[W1] I found the use of the term heterogeneous to describe the observational data a bit vague and misleading.**
>
> **Response:** We thank the reviewer for pointing out this issue. In fact, we use the term "heterogeneous" to distinguish whether the data meets the unconfoundedness assumption, i.e., $o \perp (Y_{0}, Y_{1}) \mid x$. Samples with fully observed features or from randomized controlled trial (RCT) meet such independence assumption and are denoted as $g=1$, while samples with missing features that do not satisfy the assumption are denoted as $g=0$. As suggested by the reviewer, we will clearly specify the dataset information and above difference in the revised manuscript's introduction.
>
> > **[W2] I think the related work section on causal inference (Appendix A.1) could be improved.**
>
> **Response:** We thank the reviewer for pointing out this issue and apologize for the lack of clarity. As suggested by the reviewer, below we will introduce more relevant works and discuss their limitations.
> - **Statistical test based methods** introduce statistical tests to compare the causal effects estimated from observational studies and randomized trials, thereby detecting and mitigating hidden confounding [1, 2, 3, 4]. **Correction based methods** are proposed to correct the biased causal effect estimation derived from observational data using the unbiased RCT data [5, 6, 7], and an efficient integrative estimator is established based on semi-parametric theory [8], and further integrated with machine learning models [9]. **Weighted methods** propose to train a biased estimator using observational data, train an unbiased estimator using RCT data, and take the weighted average of these two estimators as the final result [10, 11, 12].
> - A **limitation** of data fusion methods is the availability of RCT data, as the cost of obtaining RCT data is prohibitively high. Moreover, for the correction based method, the randomized trial and observational study should share the same support sets. When the support sets differ and only a partial overlap exists, additional strong parametric assumptions are required for extrapolation, for instance, the hidden confounding effect is assumed to be a linear function [5].
>
> > **[W3] The performance improvement over previous methods is not so strong.**
>
> **Response:** Thank you for the comments. In fact, we carefully tune the parameters of all baselines, so that some baseline results seem competitive. Even though, MetaDebias significantly outperforms these baselines under p-value less than 0.05. To further address your concern, we add experiments on PCIC and Ali-CCP datasets. Due to the space limit, some results are shown below, and please refer to the PDF for more details.
>
> |AUC Metric|PCIC|Ali-CCP|
> |:--:|:--:|:--:|
> |Naive|0.694±0.005|0.590±0.013|
> |DR|0.701±0.003|0.608±0.005|
> |ESCM2-DR|0.706±0.004|0.611±0.008|
> |Res-DR|0.709±0.004|0.614±0.005|
> |MetaDebias|**0.715*±0.005**|**0.626*±0.006**|
> ||
>
> The results of the statistical significance test indicate that MetaDebias significantly outperforms the baselines.
>
> > **[Minor] Typo in Lemma 1.**
>
> **Response:** We thank the reviewer for pointing out this issue, and we will carefully polish the revised paper to avoid typos and improve readability.
>
> > **[Q1] Could you explain with more details how are you assessing the statistical significance of the performance improvement over the previous methods?**
>
> **Response:** Thanks for the question. Actually, we use a paired t-test to examine whether the proposed method significantly outperform the **optimal** baseline under different metrics across various datasets. The specific testing procedure is as follows:
> - First, we conduct independent repeated experiments to calculate the mean and variance of all methods on the specified metrics.
> - Next, the t-statistic is calculated using the mean and variance from both the proposed MetaDebias and the optimal baseline results.
> - Finally, we identify the critical value based on the pre-specified degrees of freedom and significance level, and compare it with the absolute value of the t-statistic. If the absolute value of t-statistic exceeds the critical value, the difference is considered significant.
>
> For the empirical implementation, we use the ‘stats.ttest_rel’ function from the SciPy library in Python.
>
> ***
> **We sincerely thank you for your feedback and will provide more clarifications and explanations in the revised version, and welcome any further technical advice or questions on this work and we will make our best to address your concerns.**
>
> ***
> **References**
>
> [1] Hidden yet quantifiable: A lower bound for confounding strength using randomized trials
>
> [2] Falsification before Extrapolation in Causal Effect Estimation
>
> [3] Falsification of internal and external validity in observational studies via conditional moment restrictions
>
> [4] Benchmarking Observational Studies with Experimental Data under Right-Censoring
>
> [5] Removing hidden confounding by experimental grounding
>
> [6] Combining observational and randomized data for estimating heterogeneous treatment effects
>
> [7] Elastic integrative analysis of randomised trial and realworld data for treatment heterogeneity estimation
>
> [8] Improved Inference for Heterogeneous Treatment Effects Using Real-World Data Subject to Hidden Confounding
>
> [9] Integrative R-learner of heterogeneous treatment effects combining experimental and observational studies
>
> [10] Adaptive combination of randomized and observational data
>
> [11] Combining observational and experimental datasets using shrinkage estimators
>
> [12] Combining multiple observational data sources to estimate causal effects

---

> > ### Comment · Reviewer_Yf6Q · 2024-08-07
> >
> > I thank the authors for their response. In light of the clarifications and additional experiments I will raise my score.

---

> ### Author Response · Authors · 2024-08-08
> **Thank you for raising your score!**
>
> We are happy that our clarifications and additional experiments addressed your current concerns. We look forward to your continued support during the follow-up discussion period -- thank you so much!

---

### Official Review · Reviewer_X846 · 2024-07-12

**Soundness:** 4
**Presentation:** 3
**Contribution:** 4
**Rating:** 7
**Confidence:** 3

**Summary:**

The paper addresses the issue of selection bias in recommender systems, particularly when hidden confounding factors are present. The authors propose a new approach using heterogeneous observational data, where some data is affected by hidden confounding and some is not. The proposed MetaDebias is a meta-learning based debiasing method that models oracle prediction errors and bias from hidden confounders using bi-level optimization for training. Experiments on three public datasets demonstrate that MetaDebias achieves the best performance despite the presence of hidden confounding.

**Strengths:**

Addressing hidden confounders in recommender systems is a significant issue.

The approach is quite novel.

The experimental evaluation is well done, with a clear explanation of most aspects of the experimental evaluation.

The related work is broadly covered and compared to.

**Weaknesses:**

In the comparative experiments shown in Table 1, the improvement of the MetaDebias algorithm over other algorithms is relatively small.

Although the paper is well-organized, the presentation quality needs improvement. For example, some abbreviations should be spelled out when first mentioned, such as Random Controlled Trial (RCT) in the abstract.

**Questions:**

How does the proposed MetaDebias algorithm perform in terms of time and space efficiency compared to the baseline?

**Limitations:**

Yes

---

> ### Author Rebuttal · Authors · 2024-08-07
>
> We sincerely appreciate your approval of the idea and the novelty of this work, and thank you for the helpful suggestions. Below, we hope to address your concerns and questions to improve the clarity and readability of our paper.
>
> > **[W1] In the comparative experiments shown in Table 1, the improvement of the MetaDebias algorithm over other algorithms is relatively small.**
>
> **Response:** Thank you for the comments. In fact, we carefully tune the parameters of all baselines, so that some baseline results seem competitive. Even though, MetaDebias significantly outperforms these baselines under p-value less than 0.05. To further address your concern, we add experiments on PCIC and Ali-CCP datasets to validate the effectiveness of our method.
>
> - PCIC dataset contains 19420 biased ratings and 2040 unbiased ratings derived from 1000 users evaluating 1720 movies. Ali-CCP is a large-scale public dataset for click and conversion prediction in industrial scenarios, where the training, validation, and test data consist of 38 million, 4.2 million, and 43 million records, respectively.
> - The AUC performance on both datasets are shown below.
>
> |AUC Metric|PCIC|Ali-CCP|
> |:--:|:--:|:--:|
> |Naive|0.694±0.005|0.590±0.013|
> |IPS|0.696±0.004|0.602±0.004|
> |DR|0.701±0.003|0.608±0.005|
> |ESMM|0.695±0.005|0.592±0.006|
> |ESCM2-IPS|0.705±0.003|0.608±0.009|
> |ESCM2-DR|0.706±0.004|0.611±0.008|
> |Res-IPS|0.706±0.003|0.611±0.007|
> |Res-DR|0.709±0.004|0.614±0.005|
> |MetaDebias|**0.715*±0.005**|**0.626*±0.006**|
> ||
>
> The results show that the proposed MetaDebias stably outperforms the baseline methods across both datasets, and achieves a significant performance improvement on the large-scale industrial dataset Ali-CCP. Please refer to the attached PDF for more results.
>
> > **[W2] Although the paper is well-organized, the presentation quality needs improvement. For example, some abbreviations should be spelled out when first mentioned, such as Random Controlled Trial (RCT) in the abstract.**
>
> **Response:** We thank the reviewer for pointing out this issue and apologize for the lack of clarity. We will carefully polish the revised manuscript to improve readability.
>
> > **[Q1] How does the proposed MetaDebias algorithm perform in terms of time and space efficiency compared to the baseline?**
>
> **Response:** Thanks for the question. As suggested by the reviewer, we compare the parameter size, training time (minutes) and inference time (milliseconds per sample) of different methods on the Coat, Yahoo! R3 and KuaiRec datasets. The results are shown below.
>
> |KuaiRec Dataset|Parameters|Training|Inference|
> |:--:|:--:|:--:|:--:|
> |Naive|$1 \times$|91.093|0.258|
> |DR|$3 \times$|219.948|0.274|
> |TDR|$3 \times$|135.778|0.285|
> |Multi-DR|$3 \times$|238.706|0.265|
> |ESCM2-DR|$3 \times$|104.668|0.782|
> |BRD-DR|$5 \times$|198.872|0.249|
> |KD-Label|$2 \times$|163.358|0.239|
> |Autodebias|$4 \times$|204.555|0.632|
> |LTD-DR|$3 \times$| 253.277|0.267|
> |Bal-DR|$5 \times$|157.784|0.257|
> |Res-DR|$5 \times$|165.444|0.244|
> |MetaDebias|$5 \times$|174.866|0.254|
> ||
>
> |Yahoo!R3 Dataset|Parameters|Training|Inference|
> |:--:|:--:|:--:|:--:|
> |Naive|$1 \times$|0.574|0.230|
> |DR|$3 \times$|6.599|0.258|
> |TDR|$3 \times$|6.227|0.265|
> |Multi-DR|$3 \times$|5.906|0.283|
> |ESCM2-DR|$3 \times$|1.098|0.262|
> |BRD-DR|$5 \times$|2.973|0.261|
> |KD-Label|$2 \times$|2.449|0.358|
> |Autodebias|$4 \times$|1.662|0.243|
> |LTD-DR|$3 \times$| 1.832|0.262|
> |Bal-DR|$5 \times$|1.702|0.236|
> |Res-DR|$5 \times$|3.927|0.288|
> |MetaDebias|$5 \times$|6.375|0.263|
> ||
>
> |Coat Dataset|Parameters|Training|Inference|
> |:--:|:--:|:--:|:--:|
> |Naive|$1 \times$|0.206|0.250|
> |DR|$3 \times$|0.793|0.498|
> |TDR|$3 \times$|0.693|0.472|
> |Multi-DR|$3 \times$|0.552|0.515|
> |ESCM2-DR|$3 \times$|0.462|0.578|
> |BRD-DR|$5 \times$|0.682|0.347|
> |KD-Label|$2 \times$|1.124|0.519|
> |Autodebias|$4 \times$|1.182|0.395|
> |LTD-DR|$3 \times$| 2.351|0.459|
> |Bal-DR|$5 \times$|2.912|0.513|
> |Res-DR|$5 \times$|0.674|0.387|
> |MetaDebias|$5 \times$|1.916|0.521|
> ||
>
> - **Space efficiency:** For all methods, we employ a multi-layer perceptron network to model the prediction model, while the same architecture is also used for the propensity and imputation model. As shown in tables above, the Naive method employs only a single prediction model to fit the training data, with the parameter size denoted as $1 \times$, while the Doubly Robust (DR) method further incorporates both propensity and imputation model to achieve double robustness, with the parameter size denoted as $3 \times$. The parameter size of the proposed MetaDebias method is comparable to that of some existing methods such as Res-DR, indicating that the proposed method outperforms the competitive baselines under the same parameter size.
>
> - **Time efficiency:** Despite the involvement of five models in the training process, the comparison with other baseline methods reveals that the training time of the proposed approach is acceptable, particularly on the large-scale dataset KuaiRec. A potential reason for the relatively long training time required by proposed method is that the bi-level optimization process with assumed updates results in multiple gradient computations throughout the training procedure.
>
> - In summary, the computational resource demands of proposed method are acceptable. The experimental results can also be found in the one-page attached PDF.
>
> ***
> **We sincerely thank you for your feedback and will provide more clarifications and explanations in the revised version, and welcome any further technical advice or questions on this work and we will make our best to address your concerns.**

---

> > ### Comment · Reviewer_X846 · 2024-08-10
> >
> > Thank you for your response. I really like the work.
> > It will be better if the authors could provide more details about the PCIC and Ali-CCP datasets. Is PCIC public access?  Are the ratings in test set of Ali-CCP unbiased?

---

> ### Author Response · Authors · 2024-08-12
> **We are happy to provide more details about the PCIC and Ali-CCP datasets!**
>
> Thank you for engaging with our responses and we are highly encouraged to know that "you really like our work". In below, we are happy to provide more details about the **PCIC** and **Ali-CCP** datasets.
>
> - **PCIC is a public dataset** for evaluating debiasing algorithms in recommendations [1, 2]. In the training set, users are free to choose items to rate, resulting in 19,420 biased ratings. While in the test set, users are required to rate randomly exposed items, resulting to 2,040 unbiased ratings. **FYI, the PCIC dataset is public available: https://competition.huaweicloud.com/information/1000041488/introduction.**
>
> - **Ali-CCP is a public dataset** gathered from real-world traffic logs of the recommender system in an e-commerce platform [3]. The training and test set are split along the time sequence of traffic logs, which is a traditional industrial setting. Specifically, the latter 1/2 data in the time sequence are split to be test set, thus **the test data is not exactly unbiased, but with different training the inference space, which is similar to our debiased recommendation problem setup**. In summary, Ali-CCP is also a widely adopted debiased recommendation dataset especially used for **post-click conversion rate (pCVR) estimation** task in recommendation systems [3, 4, 5, 6, 7, 8]. **The Ali-CCP dataset is public available: https://tianchi.aliyun.com/datalab/dataSet.html?dataId=408.**
>
> ***
>
> We will definitely put the above experimental details and results into our revised manuscript  -- thank you so much!
>
> ***
> **References**
>
> [1] Mengyue Yang et al. Debiased Recommendation with User Feature Balancing. Transactions on Information Systems 2023.
>
> [2] Mengyue Yang et al. Generalizable Information Theoretic Causal Representation. arXiv.
>
> [3] Xiao Ma et al. Entire Space Multi-Task Model: An Effective Approach for Estimating Post-Click Conversion Rate. SIGIR 2018.
>
> [4] Wenhao Zhang et al. Large-scale Causal Approaches to Debiasing Post-click Conversion Rate Estimation with Multi-task Learning. WWW 2020.
>
> [5] Dongbo Xi et al. Modeling the Sequential Dependence among Audience Multi-step Conversions with Multi-task Learning in Targeted Display Advertising. KDD 2021.
>
> [6] Hao Wang et al. ESCM2: entire space counterfactual multi-task model for post-click conversion rate estimation. SIGIR 2022.
>
> [7] Xiaofan Liu et al. Task Adaptive Multi-learner Network for Joint CTR and CVR Estimation. WWW 2023.
>
> [8] Xinyue Zhang et al. Adversarial-Enhanced Causal Multi-Task Framework for Debiasing Post-Click Conversion Rate Estimation. WWW 2024.

---

> > ### Comment · Reviewer_X846 · 2024-08-14
> >
> > Thanks for your response. I will keep my score "7: Accept".

---

### Official Review · Reviewer_TTAY · 2024-07-18

**Soundness:** 2
**Presentation:** 2
**Contribution:** 2
**Rating:** 3
**Confidence:** 3

**Summary:**

This paper proposes to use heterogeneous observational data to address hidden confounding in recommender system.

**Strengths:**

+ Addressing selection bias in recommender system is very important.
+ If the assumption holds, i.e., confounder missing mechanism follows the user attribute missing mechanism, I would say the method makes sense to me, though it is complicated.
+ Experiments seem to demonstrate the effectiveness of the proposed method.

**Weaknesses:**

- It seems to me that using missing feature mechanism to estimate the missing confounder mechanism is a over simplification of the problem, where the author provide no strong empirical evidence to demonstrate the reliability of this simplification.
- If confounders are just missing features, why don't we just infer them from the data?

**Questions:**

Please refer to my summary of weakness.

---

> ### Author Rebuttal · Authors · 2024-08-07
>
> We sincerely appreciate the reviewer’s great efforts and insightful comments to improve our manuscript. Below, we hope to address your concerns and questions to improve the clarity and readability of our paper.
>
> > **[W1] It seems to me that using missing feature mechanism to estimate the missing confounder mechanism is a over simplification of the problem, where the author provide no strong empirical evidence to demonstrate the reliability of this simplification.**
>
> **Response:** Thank you for the comments. Below, we will demonstrate why hidden confounders are equivalent to missing features within the potential outcome framework, and introduce the motivation of our work.
> - In fact, we follow previous works and adopt the potential outcome framework to formalize the debiasing problem in the presence of hidden confounding. Specifically, we define $X$ as the features of user-item pair, such as user gender and item color, which influence both click and purchase behaviors. Here, click $T$ and purchase $Y$ are respectively defined as the treatment and outcome.
> - Notably, previous works rely on the unconfoundedness assumption (also known as ignorability), i.e., $T \perp (Y_{0}, Y_{1}) \mid X$, where $Y_{0}$ and $Y_{1}$ are potential outcomes. In causal inference, this indicates that given the observed features, the purchase outcome if the user clicks is independent of the click behavior itself. In recommendations, this indicates the observed features are sufficient to model both click and purchase behaviors.
> - However, the assumption may sometimes be violated. For example, user income is the feature that may simultaneously influence both clicks and purchases, but if it is missing, it leads to a violation of the unconfoundedness assumption. Therefore, hidden confounders are equivalent to missing features. In fact, it is a widely adopted setup in many previous studies.
> - Note that, when only observational data with hidden confounding is available, even with strong assumptions, it remains difficult to eliminate the hidden confounding bias. This illustrates why recent works propose incorporating RCT data for calibration. However, the cost of acquiring RCT data is exceptionally high due to the requirement for random assignment of treatment. In recommendation scenarios, this requires users to randomly click and rate items. This motivates us to utilize observational data with fully observed features rather than RCT data to address hidden confounding.
>
> > **[W2] If confounders are just missing features, why don't we just infer them from the data?**
>
> **Response:** Thank you for the question. Below, we will discuss the feasibility of feature imputation methods from both theoretical and experimental perspectives. In this study, we categorize the data into two groups, $g=1$ and $g=0$, based on whether they satisfy the unconfoundedness assumption. In fact, the samples with $g=1$ can be further categorized into two types.
> - As shown in the motivation graph in the paper, the first case where $g=1$ corresponds to complete feature collection. With fully observed features, it is possible to estimate the joint distribution of all features, and allows the missing feature inference as the reviewer noted. However, there may exist selection bias, meaning that the features are missing not at random, which hinders the accurate missing feature imputation.
> - Below are the results of three different feature imputation methods on three benchmark datasets. The Sample-Imp method imputes missing values by sampling from a Gaussian distribution, where the mean and variance of the distribution are estimated using the features with $g=1$. The Naive-Imp method aims to learn a model for imputing missing features from observed features by training on the naive loss on the samples with $g=1$, while IPW-Imp further incorporates propensity scores to account for selection bias.
>
> |NDCG@K|Coat|Yahoo!R3|KuaiRec|
> |:--:|:--:|:--:|:--:|
> |Naive|0.444±0.014|0.489±0.009|0.540±0.009|
> |Sample-Imp|0.449±0.012|0.498±0.012|0.545±0.006|
> |Naive-Imp|0.458±0.013|0.496±0.013|0.556±0.008|
> |IPW-Imp|0.462±0.011|0.500±0.009|0.563±0.006|
> |MetaDebias|**0.473*±0.010**|**0.544*±0.005**|**0.584*±0.003**|
> ||
>
> The results show that the feature imputation methods achieve performance improvements compared to the naive approach, but the proposed MetaDebias method still significantly outperforms all baselines. Moreover, IPW-Imp outperforms Naive-Imp, indicating that the features are missing not at random and imputing missing values remains a challenge.
>
> - RCT data satisfy the independence condition of unconfoundedness assumption, and thus is also labeled as $g=1$. Such data is obtained through the random assignment of treatment, and does not require the collection of complete features. In this case where complete features are absent as inferring labels, feature imputation methods often struggle to achieve good performance.
> - Below, we compare the feature imputation method with proposed MetaDebias, where samples with $g=1$ are all derived from randomized trials, and the Random-Imp method employs random values from [-0.5, 0.5] to impute missing feature values.
>
> |NDCG@K|Coat|Yahoo!R3|KuaiRec|
> |:--:|:--:|:--:|:--:|
> |Naive|0.450±0.012|0.498±0.015|0.545±0.006|
> |Random-Imp|0.452±0.013|0.492±0.013|0.542±0.006|
> |MetaDebias|**0.468*±0.012** |**0.536*±0.008**|**0.581*±0.003**|
> ||
>
> Experimental results indicate that when samples with $g=1$ are from randomized trials, the performance of feature imputation method is even inferior than the naive approach that does not involve imputation, and significantly worse than the proposed MetaDebias. In this case, feature imputation fails.
>
> ***
> **We hope the above discussion will fully address your concerns about our work, and we would really appreciate it if you could be generous in raising your score.** We look forward to your insightful and constructive responses to further help us improve the quality of our work. Thank you!

---

> ### Author Response · Authors · 2024-08-12
> **We would like to supplement further clarification for the equivalence of "Hidden Confounding" and "Missing Features (Covariates)".**
>
> Dear Reviewer TTAY,
>
> Thank you again for your time to review our paper and thoughtful feedback. In below, we would like to further supplement two main claims to help readers understand our problem setup.
>
> **Main Claim 1: For the potential outcome framework [1, 2] in causal inference, ```hidden confounding``` and ```missing features (covariates)``` are  equivalent, supported by the relevant literature published in important venues [3, 4, 5, 6].**
>
> - In [3], Section 2 (Binary Outcome) on page 3, the ```unmeasured confounder``` is defined as $U$, whereas in Section 3 (Survival Time) on page 8, $U$ is referred to as a ```covariate```.
>
> - In [4], on page 1, the authors explicitly wrote that _“throughout the article we use the term ```‘unmeasured confounder’``` rather than using terms such as ```‘omitted variable’``` or ```‘unobserved covariate’``` for the sake of consistency and clarity”_.
>
> - In [5], the ```unobserved covariate``` $U$ is first defined in the unconfoundedness assumption in Section 2 (Problem Statement and Preliminaries) on page 2. In the following Section 2.1 (Related Work), $U$ is explicitly referred to as the ```unobserved confounder```.
>
> - In [6], Section 2 (Preliminaries and Challenges to Identification) on page 5, the problem formulation is explicitly stated as follows, _"we are interested in the effects of $X$ on $Y$ , which may be ```confounded by a vector of $q$ unobserved covariates $U$```”_, which indicates the equivalence of confounders and covariates.
>
> **Main Claim 2: In recommendation systems, prior works addressing hidden confounding similarly builds on the equivalence between ```hidden confounding``` and ```missing features``` [7, 8, 9].**
>
> - In [7], Section 2 (Problem Formulation) on page 2, $x_{u,i}$ is defined as the ```observed features``` of user-item pair $(u, i)$, and is considered to be the ```measured confounders```. In addition, ```unmeasured confounders``` $h_{u,i}$ refer to the ```unobserved features```, as shown in Figure 2.
>
> - In [8], the authors claim that “we assume that ```all confounders consist of a measured part $x_{u,i}$ and a hidden (unmeasured) part $h_{u,i}$```, where the latter arises from issues such as information limitations (e.g., friend suggestions) and privacy restrictions (e.g., user salary)” in the Section 2 (Problem Setup) on page 3.
>
> - In [9], the authors state that “The ```observed feature/confounder``` $x_{u,i}$ refers to the ```observed feature``` vector from the user $u$, item $i$”, which indicates the consistency between the confounders and the features.
>
> From above, **we conclude that the terminologies ```hidden confounders``` and ```unobserved features``` are exactly equivalent, not over-simplification, in the potential outcome framework of causal inference and debiased recommendation literature. In addition, we also added extensive experiments to validate the superiority of our method compared to the "simple" feature imputation methods** (please kindly refer to our rebuttal). We will definitely put the above discussions and added experiments into our final version to fully address your concern!
>
> ***
>
> Could you please check whether they properly addressed your concern? If there are more remaining issues, we would appreciate the chance to address them and work towards achieving a higher score. We deeply appreciate all the insightful comments you have posted, as they have greatly enhanced our paper!
>
> With thanks and warm wishes,
>
> Submission9441 Authors
>
> ***
> **References**
>
> [1] Donald B Rubin. Estimating causal effects of treatments in randomized and nonrandomized studies. Journal of educational psychology 1974.
>
> [2] Neyman et al. On the application of probability theory to agricultural experiments. Statistical Science 1990.
>
> [3] Danyu Lin et al. Assessing the sensitivity of regression results to unmeasured confounders in observational studies. Biometrics 1998.
>
> [4] Nicole Bohme Carnegie et al. Assessing sensitivity to unmeasured confounding using a simulated potential confounder. Journal of Research on Educational Effectiveness 2016.
>
> [5] Nathan Kallus et al. Confounding-robust policy improvement. NeurIPS 2018.
>
> [6] Wang Miao et al. Identifying effects of multiple treatments in the presence of unmeasured confounding. Journal of the American Statistical Association 2023.
>
> [7] Sihao Ding et al. Addressing Unmeasured Confounder for Recommendation with Sensitivity Analysis. KDD 2022.
>
> [8] Haoxuan Li et al. Removing Hidden Confounding in Recommendation: A Unified Multi-Task Learning Approach. NeurIPS 2023.
>
> [9] Zhiheng Zhang et al. Robust causal inference for recommender system to overcome noisy confounders. SIGIR 2023.

---

> > ### Author Response · Authors · 2024-08-14
> >
> > Dear Reviewer TTAY,
> >
> > Since the discussion period will end in a few hours, we will be online waiting for your feedback on our rebuttal, which we believe has fully addressed your concerns.
> >
> > We would highly appreciate it if you could take into account our response when updating the rating and having discussions with AC and other reviewers.
> >
> > Thank you so much for your time and efforts. Your feedback would be extremely helpful to us. If you have further comments or questions, we hope for the opportunity to respond to them.
> >
> > Many thanks,
> >
> > Submission9441 Authors

---

### Author Rebuttal · Authors · 2024-08-07

Dear reviewers and AC,

We sincerely thank all reviewers and AC for your great effort and constructive comments on our manuscript. During the rebuttal period, we have been focusing on these beneficial suggestions from the reviewers and doing our best to add several experiments.

As reviewers highlighted, we believe our paper tackles an important and relevant problem (**Reviewer TTAY**, **Reviewer X846**), and introduces a novel and interesting idea (**Reviewer X846**). We also appreciate that the reviewers found our paper well-organized (**Reviewer X846**) and offers solid and convincing experiments (**Reviewer TTAY**, **Reviewer X846**, **Reviewer Yf6Q**).

Moreover, we thank the reviewers for the suggestions for incorporating feature imputation methods (**Reviewer TTAY**), as well as pointing out the concerns regarding the empirical performance improvement (**Reviewer X846**, **Reviewer Yf6Q**), and time and space efficiency (**Reviewer X846**). In response to these comments, we have added the following experiments:

- [Reviewer TTAY] We **add experiments to compare the feature imputation methods with proposed MetaDebias** (in Table 1 and Table 2).
- [Reviewer X846 and Reviewer Yf6Q] We **add experiments on additional PCIC and Ali-CCP datasets to further validate the effectiveness** of MetaDebias (in Table 3).
- [Reviewer X846] We **add experiments to investigate the time and space efficiency** on the **Coat**, **Yahoo! R3**, and **KuaiRec** datasets (in Table 4).

We hope our response could address all the reviewers' concerns, and are more than eager to have further discussions with the reviewers in response to these revisions.

Thanks,

Submission9441 Authors.

---

### Decision · Program_Chairs · 2024-09-25

**Decision:**

Accept (poster)

**Comment:**

The manuscript tackles the hidden confounder issue in offline learning from observational datasets. The key idea is to leverage heterogenous observational data (i.e., all features are completely observable in some instances) to perform error imputation, which leads to improved learning of prediction models.

The reviewers generally appreciate the idea proposed in this paper, but also expressed serious concerns regarding the clarity about the problem setup and proposed methodology. First of all, even during the rebuttal period, the authors kept emphasizing “missing features” is equivalent to “hidden/missing confounders”. This is only true under the key assumption made in this paper, which they dodged, is that there are training samples where all confounders are observed. This is clearly a very strong assumption and can hardly hold in the targeted application of this paper, i.e., recommender systems. For example, a shopping website might never observe their users’ latest salary figures or medical history.

Second, the notion of $x_{u,i}$ is quite confusing: it actually has different dimensions when $g=1$ vs $g=0$. Then it is a bit hard to imagine how to implement those prediction models, e.g., $f(x_{u,i})$ and $\eta(x_{u,i})$. We urge the authors to better explain this in the revision.

Last but not least, although the authors kept emphasizing the goal is to perform unbiased learning from observational data, there is no discussion about how or whether the proposed method could be unbiased.

To summarize, we believe this manuscript has its unique value and would be a good add to the research in unbiased learning from observational data. We urge the authors improve the clarity of the manuscript and include the discussions suggested by the reviewers.